# Incorporating Sum Constraints into Multitask Gaussian Processes

**Philipp Pilar**                                                *philipp.pilar@it.uu.se*
*Department of Information Technology*
*Uppsala University*

**Carl Jidling**                                                  *carl.jidling@it.uu.se*
*Department of Information Technology*
*Uppsala University*

**Thomas B. Schön**                                          *thomas.schon@it.uu.se*
*Department of Information Technology*
*Uppsala University*

**Niklas Wahlström**                                         *niklas.wahlstrom@it.uu.se*
*Department of Information Technology*
*Uppsala University*

**Reviewed on OpenReview:** *https://openreview.net/forum?id=gzu4ZbBY7S*

## Abstract

Machine learning models can be improved by adapting them to respect existing background knowledge. In this paper we consider multitask Gaussian processes, with background knowledge in the form of constraints that require a specific sum of the outputs to be constant. This is achieved by conditioning the prior distribution on the constraint fulfillment. The approach allows for both linear and nonlinear constraints. We demonstrate that the constraints are fulfilled with high precision and that the construction can improve the overall prediction accuracy as compared to the standard Gaussian process.

## 1 Introduction

Many real world problems come with background knowledge known a priori, for instance that the outputs must be positive at all times or fulfill a certain differential equation. The constraints are often known to near perfect precision. Any model would certainly benefit from having such knowledge hardcoded in advance instead of having to rediscover it, as the additional information would allow for the exclusion of the majority of possible outputs.

In this work we consider the Gaussian process (GP) (Rasmussen & Williams, 2006), which is a popular and powerful machine learning model. Some assumptions about the underlying function, e.g. regarding its smoothness, can be encoded in a relatively straightforward way into the kernel of the GP. However, it is usually trickier to include more specific prior knowledge and constrained GPs (or, for that matter, constrained machine learning methods) constitute a relevant and active area of research (Willard et al., 2021; Swiler et al., 2020).

In this work, we focus on constraints that take the form of a sum over the outputs of a multitask GP. Constraints of this form arise, for example, when considering conserved quantities in physics such as energy and momentum, where the sum over the energies or momenta of all subcomponents of a closed system must remain constant. As a toy example, we consider the harmonic oscillator, which is ubiquitous in physics; the

expression for the energy takes the form

$$E = E_{\text{pot}}(t) + E_{\text{kin}}(t) = kz(t)^2/2 + mv(t)^2/2, \tag{1}$$

where $E_{\text{pot}}$ and $E_{\text{kin}}$ denote potential and kinetic energy, respectively. We assume that the displacement from the rest position $z$ and the velocity $v$ are the outputs of a multitask GP, whereas the time $t$ serves as input. While the input in this example is one-dimensional, the results we derive in this paper also apply to higher dimensional inputs.

We have developed a method that allows nonlinear constraints like (1) to be incorporated into the GP. First, we show how nonlinear constraints can be reduced to linear ones via a suitable transformation of the outputs of the GP. Then we proceed to condition the joint prior of the GP on the constraints, which in turn results in a constrained predictive distribution. In the next section, we start by providing a formal definition of the problem.

## 2 Problem Formulation

### 2.1 Background on the GP

A GP is formally defined as "a collection of random variables, any finite number of which have a joint Gaussian distribution" (Rasmussen & Williams, 2006). Formally, we write $f(\mathbf{x}) \sim \mathcal{GP}(m(\mathbf{x}), k(\mathbf{x}, \mathbf{x}'))$, where $m(\mathbf{x}) = \mathbb{E}[f(\mathbf{x})]$ and $k(\mathbf{x}, \mathbf{x}') = \mathbb{E}[(f(\mathbf{x}) - m(\mathbf{x}))(f(\mathbf{x}') - m(\mathbf{x}'))]$ are the mean and the covariance function of the GP, respectively. The dataset available for training the GP consists of inputs $\mathbf{X} = \{\mathbf{x}_k\}_{k=1}^N$ and noisy outputs $y_k = f(\mathbf{x}_k) + \epsilon_k$, where we assume Gaussian noise $\epsilon_k \sim \mathcal{N}(0, \sigma_n^2)$. We use $\mathbf{y}$ to denote a vector storing all $N$ outputs.

In the following we consider the multitask setting (Bonilla et al., 2008; Skolidis & Sanguinetti, 2011), where a vector $\mathbf{f}(\mathbf{x})$ of $N_f$ outputs is learned. The overall GP framework remains unchanged but the output vector $\mathbf{f}$ (and observation vector $\mathbf{y}$) has to be interpreted as an extended vector consisting of the concatenated multitask outputs $\mathbf{f}_k = \mathbf{f}(\mathbf{x}_k)$ — that is $\mathbf{f} = [\mathbf{f}_1^\mathsf{T}, \mathbf{f}_2^\mathsf{T}, \ldots, \mathbf{f}_N^\mathsf{T}]^\mathsf{T}$ for which it holds that $\mathbf{f} \sim \mathcal{N}(\mathbf{m_f}(\mathbf{X}), \mathbf{K_{f,f'}}(\mathbf{X}, \mathbf{X}'))$.

When constructing the mean and covariance function, the different tasks need to be taken into account (Alvarez et al., 2012). We write the mean as

$$\mathbf{m_f}(\mathbf{X}) = [m_d(\mathbf{x}_1)\mathbf{m_t}(\mathbf{x}_1)^\mathsf{T}, \ldots, m_d(\mathbf{x}_N)\mathbf{m_t}(\mathbf{x}_N)^\mathsf{T}]^\mathsf{T}, \tag{2}$$

where $m_d(\cdot)$ is the data mean and $\mathbf{m_t}(\cdot)$ is the task mean. The task mean returns a column vector of length $N_f$. The covariance matrix becomes

$$\mathbf{K_{f,f'}}(\mathbf{X}, \mathbf{X}) = \begin{bmatrix} k_{d11}\mathbf{k_t}(\mathbf{x}_1, \mathbf{x}_1) & k_{d12}\mathbf{k_t}(\mathbf{x}_1, \mathbf{x}_2) & \ldots \\ k_{d21}\mathbf{k_t}(\mathbf{x}_2, \mathbf{x}_1) & k_{d22}\mathbf{k_t}(\mathbf{x}_2, \mathbf{x}_2) & \ldots \\ \vdots & \vdots & \ddots \end{bmatrix}, \tag{3}$$

where $k_{dij} = k_d(\mathbf{x}_i, \mathbf{x}_j)$, and where $k_d(\cdot, \cdot)$ and $\mathbf{k_t}(\cdot, \cdot)$ denote the data and task kernels, respectively. Note that the task kernel returns a matrix of size $(N_f, N_f)$.

The task mean and kernel are often assumed to be position independent (although this assumption is not necessary for our method to work); then $\mathbf{m_f}$ and $\mathbf{K_{f,f'}}$ can be written as Kronecker products

$$\mathbf{m_f}(\mathbf{X}) = m_d(\mathbf{X}) \otimes \mathbf{m_t}, \tag{4a}$$

$$\mathbf{K_{f,f'}}(\mathbf{X}, \mathbf{X}') = k_d(\mathbf{X}, \mathbf{X}') \otimes \mathbf{\Sigma_t}. \tag{4b}$$

Given the expressions for the mean and the kernel, the predictive distribution is formed through the standard procedure; see Section B.1 in the supplementary material for details. See also Section B.2, for details on how to deal with the case of incomplete measurements, i.e. when there are data points $\mathbf{y_k}$ for which only some of the output tasks have been measured.

### 2.2  Sum Constraint

The main concern of this work is to show how constraints on the sum of some (nonlinear) transformations $h_i(\cdot)$ of the outputs $f_i$ can be incorporated into the GP. Formally, we define this class of constraints as

$$\mathcal{F}[\mathbf{f}(\mathbf{x})] = \sum_i a_i(\mathbf{x}) h_i(f_i(\mathbf{x})) = C(\mathbf{x}), \tag{5}$$

where the functions $a_i(\mathbf{x})$ serve as prefactors to the various terms in the sum, $i$ indexes the outputs of the GP, and $C(\mathbf{x})$ specifies what value the sum over the outputs should equal at position $\mathbf{x}$ in the input space. In the following we refer to constraints of this form as *sum constraint*.

In the general case (5), we consider input-dependent constraints $C(\mathbf{x})$ and $a_i(\mathbf{x})$. This requires knowledge of the functions $C(\mathbf{x})$ and $a_i(\mathbf{x})$, which could be practically infeasible. Hence, an important special case of (5) is the *constant sum constraint*

$$\mathcal{F}[\mathbf{f}(\mathbf{x})] = \sum_i a_i h_i(f_i(\mathbf{x})) = C, \tag{6}$$

with constant prefactors $a_i$ and constant sum $C$.

One example of a constant sum constraint is the previously mentioned energy conservation for the harmonic oscillator (1). There we have $a_1 = k/2$, $a_2 = m/2$, $h_1(z) = z^2$, $h_2(v) = v^2$ and $C = E$. Other situations where sum constraints arise include learning of probabilities that must sum to one, and the case of mechanical equilibrium where the sum of acting forces must be zero at each point.

## 3  Method

Let us now develop the methodology required to incorporate sum constraints as defined in Section 2.2 into the GP. In Section 3.1.1, we consider the case where all the outputs of the GP enter the sum constraint via a monotonic (invertible) nonlinearity and show how to reduce it to a linear sum constraint. In Section 3.1.2 we extend the procedure to sum constraints with non-monotonic nonlinearities. Finally, we show in Section 3.2 how to include linear sum constraints into the GP and hence, via the aforementioned reductions, also nonlinear sum constraints.

### 3.1  Reduction to Linear Constraint

### 3.1.1  Monotonically Increasing Nonlinearity

Consider the sum constraint (5) — while the constraint is nonlinear in terms of the outputs, it is linear in terms of the transformed outputs $h_i(f_i)$; defining $f_i' = h_i(f_i)$ and substituting it into (5) yields

$$\mathcal{F}[\mathbf{f}'(\mathbf{x})] = \sum_i a_i(\mathbf{x}) f_i'(\mathbf{x}) = C(\mathbf{x}), \tag{7}$$

which is linear in the transformed outputs $f_i'$. Hence, we can train a GP to predict the transformed outputs obeying the linear constraint (7) and backtransform to the original outputs via $f_i = h_i^{-1}(f_i')$. Note that this GP needs to be trained on transformed data $\mathbf{y}'$, where $y_i' = h_i(y_i)$. This approach requires that the nonlinear functions $h_i(\cdot)$ are invertible, otherwise it is not possible to unambiguously recover the $f_i$. See also Snelson et al. (2004).

However, it is not necessary for $h_i$ to be invertible on its entire domain. Consider the case where it is known that the output $f_i$ is restricted to an invertible subregion of the domain of $h_i$; then we solve the problem by choosing the backtransformation $h_i^{-1}$ corresponding to this subregion. For example, in case of the square function, we can consider the case where $f_i$ is known to be always positive (or always negative). Then we can just restrict the domain of the nonlinearity $h_i$ to the positive (negative) half-axis where the function is in fact invertible.

When employing the transformation (7), it is important to keep in mind that the GP prior now has to be chosen in a way suitable for the transformed outputs $f'$ instead of $f$; depending on the transformations $h(\cdot)$

---

**Algorithm 1** The Constrained GP: High-level Procedure

    **Step 1:** train an unconstrained GP on the data $\mathbf{y}$ to obtain the auxiliary outputs $\mathbf{f}_{\mathrm{aux}}$
          - (optional) use the posterior mean of $\mathbf{f}_{\mathrm{aux}}$ to create virtual measurements
    **Step 2:** train the constrained GP on the transformed data $\mathbf{y}'$ (for details, see Algorithm 2) to obtain $\mathbf{f}'$
          - (optional) (re)learn the auxiliary outputs together with the constrained outputs
    **Step 3:** backtransform the transformed outputs $\mathbf{f}'$ using the posterior mean of $\mathbf{f}_{\mathrm{aux}}$ from Step 1

---

involved, this could prove to be more challenging. We recover credible intervals for $f$ in the same way as we recover $f$, by backtransforming them; for more details, see Section B.7 in the Supplementary material.

Furthermore, the noise corresponding to the transformed data $\mathbf{y}'$ will in general not be normally distributed anymore, which means that GP regression loses its analytical tractability due to the resulting non-Gaussian likelihood. Methods to deal with non-Gaussian likelihoods include the Laplace approximation (Williams & Barber, 1998; Vanhatalo et al., 2009), variational inference (Blei et al., 2017; Tran et al., 2016), and expectation propagation (Minka, 2001). Due to its simplicity, in this work we use the Laplace approximation to deal with this issue, where applicable. It enables us to approximate non-Gaussian distributions with a Gaussian; see Appendix B.3 for details.

### 3.1.2 Non-monotonically Increasing Nonlinearity

In the previous section we showed how to reduce nonlinear sum constraints to linear ones, as long as the nonlinearities are monotonic. However, this is a rather limiting assumption as it would exclude e.g. the square function $h(f) = f^2$ from the admissible transformations. Here we describe a way of circumventing this problem.

The idea underlying our solution is to introduce one (or multiple) auxiliary variables that allow for a unique backtransformation. Typically, the auxiliary variables will keep track of where in the domain of $h(\cdot)$ it is that $f'$ lies, such that the correct local inverse can be chosen when backtransforming. In case of the square function, we can add the auxiliary output $f_{\mathrm{aux}} = f$ and retrieve the initial output $f$ via $f = \mathrm{sign}(f_{\mathrm{aux}})h^{-1}(f') = \mathrm{sign}(f_{\mathrm{aux}})\sqrt{h(f)}$. While the initial output $f$ is a practical choice here, this is in general not necessary and $f_{\mathrm{aux}}$ can be chosen arbitrarily.

There is no guarantee that learned values $f'$ will always fall within the domain of the backtransformation. If it happens that a predicted value lies outside, a pragmatic solution is to approximate $f'$ with the closest valid value; for example zero in case of negative valued predictions for square values.

Sometimes more information can be extracted from $f_{\mathrm{aux}}$ and used to ameliorate the transformed data $y'$, for instance when the backtransformation switches from one local inverse to another; then we can add virtual measurements for $f'$ at those points and force the constrained GP towards values consistent with $f_{\mathrm{aux}}$, which can significantly reduce artefacts in the backtransformed outputs $f$. Note that this can come at the cost of overconfident credible intervals in the vicinity of the virtual measurements.

In Algorithm 1, we summarize this procedure. In most cases, it is advantageous to learn the auxiliary outputs in a separate GP in Step 1, independently of the constrained outputs; when virtual measurements are to be created, this is required. Optionally, auxiliary outputs can be (re)learned in Step 2; for some examples, this can stabilize the hyperparameter learning of the constrained GP. However, when virtual measurements are involved, the prediction $\mathbf{f}_{\mathrm{aux}}$ from Step 1 should also be used for the backtransformation.

We illustrate the approach by returning to the harmonic oscillator (1), with the transformed outputs $f'_1 = z^2$ and $f'_2 = v^2$ (see also the last paragraph in Section 2.2). We choose the auxiliary outputs as $f^1_{\mathrm{aux}} = z$ and $f^2_{\mathrm{aux}} = v$, which we use to extract the sign when backtransforming $f'_1$ and $f'_2$; furthermore, we use the auxiliary outputs to create virtual measurements for $f'_1$ and $f'_2$ at the zero crossings of the posterior mean of $f^1_{\mathrm{aux}}$ and $f^2_{\mathrm{aux}}$. In order to fit the transformed outputs of the GP, the observations $\mathbf{y}_k = [z_k, v_k]^\mathsf{T}$ are transformed analogously to obtain $\mathbf{y}'_k = [z^2_k, v^2_k, z_k, v_k]^\mathsf{T}$; $z_k$ and $v_k$ are part of $\mathbf{y}'_k$ since we chose to relearn them together with the constrained outputs to improve the performance. The virtual measurements are also included in the transformed data $\mathbf{y}'$. In terms of the transformed outputs the constraint can be written compactly as

---

**Algorithm 2** Constraining the GP (Section B.1 refers to the Supplementary material)

---

**Input:** mean $\mathbf{m_f}(\cdot)$; kernel $\mathbf{K_{f,f'}}(\cdot,\cdot)$; constraints $(\mathbf{F}, \mathbf{S})$; (transformed) data $\mathbf{X}, \mathbf{y'}$; points of prediction $\mathbf{X}_*$
**Output:** constrained predictive distribution $\mathbf{f}'_*|\mathbf{X}, \mathbf{y'}, \mathbf{X}_*$
**Note:** During hyperparameter optimization $\mathbf{X}_* = \{\}$ and hence $\mathbf{f}'_* = \{\}$
**Step 1:** Construct the joint prior distribution for $[\mathbf{f}', \mathbf{f}'_*]^\mathsf{T} \sim \mathcal{N}(\boldsymbol{\mu}_0, \boldsymbol{\Sigma}_0)$ according to (B.1)
        -omit noise term $\sigma_n^2 \mathbf{I}$
**Step 2:** Construct $\mathbf{F}_{\text{tot}}$, $\mathbf{S}_{\text{tot}}$ according to (9b)
**Step 3:** Use $\mathbf{F}_{\text{tot}}$, $\mathbf{S}_{\text{tot}}$ to calculate constrained $\boldsymbol{\mu}', \boldsymbol{\Sigma}'$ according to (8b)
**Step 4:** Remove entries in $\boldsymbol{\mu}'$, $\boldsymbol{\Sigma}'$ corresponding to incomplete measurements as detailed in Section B.2
**if** *Hyperparameter optimization* **then**
    **Step 5:** Calculate the log marginal likelihood according to (B.7c)
    **Step 6:** Perform optimization step
**else if** *Prediction* **then**
    **Step 5:** Calculate the predictive distribution $\mathbf{f}'_*|\mathbf{X}, \mathbf{y'}, \mathbf{X}_*$ according to (B.7a)
**end if**

---

$\mathbf{Ff}' = C$, where $\mathbf{F} = [a_1, a_2, 0, 0]$. For more details on the harmonic oscillator dataset, see Section C.1 in the supplementary material.

### 3.2 Solving with Linear Constraints

Having shown how to reduce nonlinear sum constraints to linear ones, we proceed to describe how to incorporate linear sum constraints into the GP. The idea is to make use of the fact that sampling from a GP is equivalent to sampling from a multivariate Gaussian distribution, where the mean and covariance are obtained by evaluating the mean and the kernel of the GP at the points of interest.

Let the random vector $\mathbf{f}' \sim \mathcal{N}(\boldsymbol{\mu}, \boldsymbol{\Sigma})$; we are interested in the conditional distribution $\mathbf{f}'| \sum_i a_i f'_i = C$. More generally, to include multiple sum constraints, we want to find the distribution $\mathbf{f}'|\mathbf{Ff}' = \mathbf{S}$, where the rows of the matrix $\mathbf{F}$ contain the coefficients for each of the $N_F$ sum constraints to be included, and the elements of the vector $\mathbf{S}$ contain the corresponding sums; compare equation (9a) below.

The required conditional distribution can be calculated analytically (Majumdar & Majumdar, 2019) as

$$(\mathbf{f}'|\mathbf{Ff}' = \mathbf{S}) \sim \mathcal{N}(\boldsymbol{\mu}', \boldsymbol{\Sigma}'), \tag{8a}$$

where

$$\begin{aligned} \boldsymbol{\mu}' &= \mathbf{A}\boldsymbol{\mu} + \mathbf{D}^\mathsf{T}\mathbf{S}, \qquad \boldsymbol{\Sigma}' = \mathbf{A}^\mathsf{T}\boldsymbol{\Sigma}\mathbf{A}, \\ \mathbf{D} &= (\mathbf{F}\boldsymbol{\Sigma}\mathbf{F}^\mathsf{T})^{-1}\mathbf{F}\boldsymbol{\Sigma}^\mathsf{T}, \ \mathbf{A} = \mathbf{I}_n - \mathbf{D}^\mathsf{T}\mathbf{F}. \end{aligned} \tag{8b}$$

Of course, we need to enforce the constraint at all $N_{\text{tot}}$ data points — to that end, we construct the blockdiagonal matrix $\mathbf{F}_{\text{tot}}$ and the vector $\mathbf{S}_{\text{tot}}$ according to

$$\mathbf{F}(\mathbf{x}) = \begin{bmatrix} a_1(\mathbf{x}) & a_2(\mathbf{x}) & \dots \\ b_1(\mathbf{x}) & b_2(\mathbf{x}) & \dots \\ \vdots & \vdots & \end{bmatrix}, \qquad\qquad \mathbf{S}(\mathbf{x}) = \begin{bmatrix} C_a(\mathbf{x}) \\ C_b(\mathbf{x}) \\ \vdots \end{bmatrix}, \tag{9a}$$

$$\mathbf{F}_{\text{tot}} = \text{diag}(\mathbf{F}(\mathbf{x_1}), \mathbf{F}(\mathbf{x_2}), \dots), \qquad\qquad \mathbf{S}_{\text{tot}} = [\mathbf{S}(\mathbf{x_1})^\mathsf{T}, \dots]^\mathsf{T}. \tag{9b}$$

We use $N_{\text{tot}}$ in two different contexts: during the hyperparameter optimization, $N_{\text{tot}}$ denotes the number of data points; whereas during prediction, $N_{\text{tot}}$ denotes the number of both data and predictive points.

Algorithm 2 summarizes the practical procedure of constructing the covariance and the mean, both during hyperparameter optimization and when forming the constrained predictive distribution of the GP. In case of a position dependent constraint, it is important to note that the values of the functions $C(\mathbf{x})$ and $a_i(\mathbf{x})$ must be known at all points for which the constraint should be enforced; in our case, this means all $N_{\text{tot}}$ points.

Note that in Step 1 of the algorithm we first omit the noise term, since the constraints only hold exactly for noiseless data; the noise then enters in Step 4, after the constraints have been taken into account.

Mathematically, the constraint is enforced by conditioning the Gaussian distribution on it. While the method is not strictly global in the sense of providing a constrained kernel for the GP, it is global for practical purposes as the constraint is enforced at all points of prediction of the GP.

Due to the matrix inversion in (8b), the computational complexity of the algorithm is cubic with leading order term $\sim \mathcal{O}(N_F^3 N_{\text{tot}}^3)$, during both hyperparameter optimization and prediction.

### 3.2.1 Special Case of Constant Constraints

In the special case of constant constraints and constant inter-task dependencies of the GP mean and kernel, the constraints can be incorporated more efficiently. Here, the kernel of the GP factorizes into data and task kernel as in (4) and the procedure above simplifies: it now suffices to enforce the constraints $(\mathbf{F}, \mathbf{S})$ on the task mean and covariance matrix and to subsequently perform the Kronecker product with the data mean and covariance matrix to obtain the constrained distribution.

Formally, this can be written as follows: let $\boldsymbol{\mu_t}$ and $\boldsymbol{\Sigma_t}$ be the task mean and covariance matrix, respectively; then the constrained quantities $\boldsymbol{\mu_t'}$ and $\boldsymbol{\Sigma_t'}$ are calculated via (8b), using $\mathbf{F}$ and $\mathbf{S}$ (since the task mean and covariance matrix are constrained directly, it is not necessary to construct $\mathbf{F}_{\text{tot}}$ and $\mathbf{S}_{\text{tot}}$). Finally, the full constrained mean and covariance matrix are constructed via $\boldsymbol{\mu'} = \mathbf{m} \otimes \boldsymbol{\mu_t'}$ and $\boldsymbol{\Sigma'} = \mathbf{K} \otimes \boldsymbol{\Sigma_t'}$, where $\mathbf{m}$ and $\mathbf{K}$ are the data mean and covariance matrix, respectively. Due to the constant constraint, the data mean is also required to be constant. Without loss of generality, we choose it as $\mathbf{m} = \mathbf{1}_{N_{\text{tot}}}$ (compare B.6.1). This procedure is summarized in Algorithm 3 in the Supplementary material. Furthermore, we provide proof that this approach is indeed equivalent to the more general approach from Section 3.2 in Appendix B.6.1.

Now, the complexity of the matrix inversion involved in (8b) is reduced to $\sim \mathcal{O}(N_F^3)$; since $\boldsymbol{\mu_t}$ and $\boldsymbol{\Sigma_t}$ are constrained directly it no longer depends on $N_{\text{tot}}$ (compare also (4)). This constitutes a significant improvement over the general algorithm as usually $N_F \leq N_f \ll N_{\text{tot}}$, where $N_F$ is the number of constraints and $N_f$ the number of tasks. Whenever applicable, it is preferable to use this way of incorporating the constraint, since it is more efficient and numerically more stable than the general procedure given in Algorithm 2.

## 4 Experimental Results

In this section, we demonstrate our method at the hand of two simulation experiments and one real data experiment [1]. They have in common that the constraints involved are constant (see Section 3.2.1); for examples of the non-constant case, see Sections A.2 and A.3 in the Supplementary material.

### 4.1 Toy Problem Revisited

We gave a formulation of the auxiliary variables approach for the harmonic oscillator in Section 3.1.2 and detailed information on the dataset can be found in Section C.1 in the Supplementary material. Figure 1 illustrates this approach. The constrained GP achieves higher overall accuracy around extremal points, where the prediction is more robust with regard to the influence of random noise. In addition, the constrained GP manages to mitigate the negative effect of incomplete measurements, i.e. data points where only one of the two output dimensions has been measured, better than the unconstrained one (compare left part of $v_{\text{aux}}$ in the figure). This is natural, since the constrained GP has implicitly added a correlation between the two outputs, which the unconstrained GP is lacking.

The credible intervals in Figure 1 clarify another advantage of the constrained GP: when multiple outputs are learned to a different degree of certainty, information can be transferred from high- to low-credibility outputs, thereby narrowing the credible intervals also for the latter. This is clearly visible in areas with incomplete measurements. On the other hand, credible intervals tend to be overconfident in the vicinity of

---

[1]The code used for the experiments is available at https://github.com/ppilar/SumConstraint.

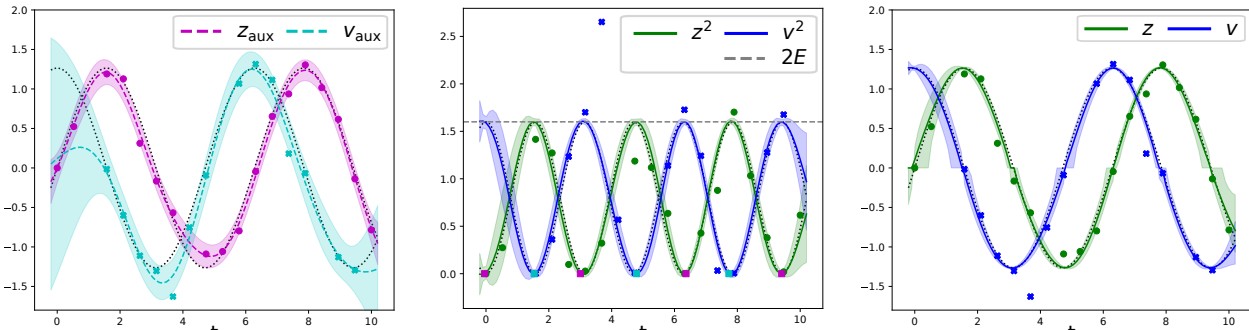

Figure 1: Demonstration of the auxiliary variables approach for the harmonic oscillator. The quantities $z_{\mathrm{aux}}$, $z$ and $v_{\mathrm{aux}}$, $v$ refer to the position and velocity of the harmonic oscillator, respectively. We distinguish between $z_{\mathrm{aux}}$, $z$ and $v_{\mathrm{aux}}$, $v$ to emphasize that, while they aim to approximate the same curve, they are learned by different GPs. The posterior means of the GPs are depicted, together with the $2\sigma$ credible intervals. The dotted lines represent the true curves and the big dots/crosses correspond to the data available to the GPs. **Left:** Results for the unconstrained GP are shown. For this example, these outputs coincide with the auxiliary outputs required for the constrained GP. **Middle:** The transformed outputs learned by the constrained GP are depicted, together with the constraint $2E = kz^2 + mv^2$ (where $k = m = 1$). The results for the auxiliary outputs have been employed to create virtual measurements at zero crossings (differently colored squares) in order to force the quadratic functions towards zero. **Right:** The backtransformed outputs of the constrained GP are shown, where the auxiliary outputs $z_{\mathrm{aux}}$ and $v_{\mathrm{aux}}$ have been used to recover the signs.

|  |  | $\sigma_n = 0.05$ | | $\sigma_n = 0.1$ | | $\sigma_n = 0.3$ | | |
|---|---|---|---|---|---|---|---|---|
|  |  | GP-c | GP-u | GP-c | GP-u | GP-c | GP-u | |
| $f_d = 0$ | RMSE | **2.3**±0.6 | 3.2±0.5 | **4.4**±1.2 | 6.4±1.1 | **13.7**±3.7 | 18.5±3.3 | (e-2) |
|  | $\lvert\Delta C\rvert$ | **0.0**±0.0 | 3.1±0.7 | **0.0**±0.0 | 6.5±1.4 | **0.1**±0.1 | 18.7±4.4 | (e-2) |
| $f_d = 0.2$ | RMSE | **3.4**±3.0 | 4.8±2.8 | **5.4**±1.9 | 8.0±2.5 | **17.0**±5.9 | 23.0±5.7 | (e-2) |
|  | $\lvert\Delta C\rvert$ | **0.0**±0.1 | 4.3±1.5 | **0.1**±0.2 | 7.8±2.2 | **0.2**±0.4 | 22.6±6.3 | (e-2) |

Table 1: Comparison of the performance of the constrained GP (GP-c) and the unconstrained GP (GP-u) for the harmonic oscillator. Shown are the root mean squared error (RMSE) of the prediction as well as the mean absolute violation of the constraint, $\lvert\Delta C\rvert$. The standard deviation of the noise is given by $\sigma_n$ whereas $f_d$ is the probability with which output components have been omitted at random from the data. The values have been obtained by averaging over 50 datasets and are given plus-or-minus one standard deviation. Bold font highlights best performance.

virtual measurements. Due to the nonlinear, piecewise backtransformation, some discontinuities have been introduced in the credible intervals of the constrained GP near the zero crossings.

In Table 1, values for both the root mean squared error (RMSE) and the average absolute violation of the constraint $\lvert\Delta C\rvert$ are given for various noise levels $\sigma_n$, both with complete and incomplete measurements; in case of incomplete measurements, the output components have been omitted at random with probability $f_d = 0.2$. The values have been obtained by averaging over 50 datasets. We observe that the constrained GP fulfills the constraint with up to two orders of magnitude higher accuracy and also performs slightly better in terms of RMSE.

The reason why the constraint is not fulfilled with yet higher accuracy for the constrained GP is that around zero crossings it can occur that invalid values are predicted by the constrained GP (that is, negative values for $z^2$ and $v^2$), which we pragmatically put to zero. This is also the origin of the small artefacts visible in that region of the mean curves in the right plot of Fig. 1.

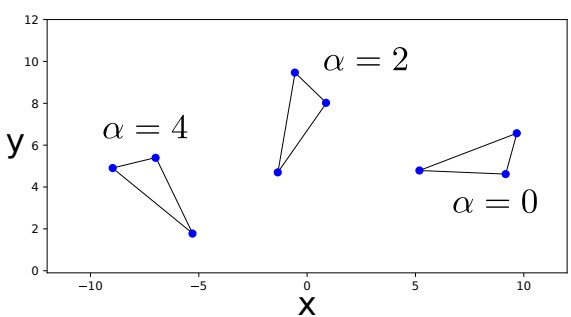

Figure 2: Visualization of the triangle in the plane. The task for the GP is to give the location of the corners of the triangle (blue dots) when given the parameter $\alpha$, which parameterizes different poses of the triangle.

| $\sigma_n$ | | GP-c | GP-u | GP-tr | |
|---|---|---|---|---|---|
| 1e-4 | RMSE | **3.3±0.2** | 4.8±0.3 | 14±30 | (e-3) |
| | $\|\Delta C\|$ | **0.3±0.0** | 1.9±0.1 | 1.8±2.7 | (e-3) |
| 1e-3 | RMSE | 5.5±1.0 | **5.0±0.3** | 9.2±16 | (e-3) |
| | $\|\Delta C\|$ | **0.8±0.1** | 2.0±0.2 | 1.6±1.3 | (e-3) |
| 1e-2 | RMSE | 4.2±0.8 | **1.6±0.2** | 3.9±0.9 | (e-2) |
| | $\|\Delta C\|$ | **6.2±1.0** | 6.4±1.4 | 8.6±1.4 | (e-3) |

Table 2: Results for the length constraint applied to the triangle in the plane. We compare results for the constrained GP (GP-c), the unconstrained GP (GP-u) and the unconstrained GP trained on the transformed outputs (GP-tr) (Salzmann & Urtasun, 2010a). For small values of noise $\sigma_n$, the sum constraint improves the performance of the GP. The values have been obtained by averaging over 50 datasets and are given plus-or-minus one standard deviation.

### 4.2 Pose Estimation

Here we demonstrate how our approach can incorporate length constraints (Perriollat et al., 2011), inspired by applications such as pose estimation. In essence, the length constraint states that the distance $L_{lm}$ between two adjacent points (indexed by $l$ and $m$) in a rigid body is constant, irrespective of position and orientation of the body. When the position is given in terms of Cartesian coordinates $z_i$, the length constraint takes the following form

$$\sum_{i=1}^{3} z_{li}^2 - 2z_{li}z_{mi} + z_{mi}^2 = L_{lm}^2. \tag{10}$$

This constraint is no longer an instance of the sum constraint as defined in (5), since the middle term depends on multiple outputs. However, with a more elaborate transformation procedure, the sum constraint can still be applied.

To make this more concrete, we consider the example of a triangle in the plane. Here, the outputs of interest are the coordinates of the triangle corners, $\mathbf{f} = [z_{1x}, z_{1y}, z_{2x}, z_{2y}, z_{3x}, z_{3y}]$. The input $\alpha$ is a continuous parametrization of different poses of the triangle in the plane. Although $\alpha$ is one-dimensional in this example, the approach generalizes to higher dimensional inputs. In our choice of transformed outputs, we follow the approach by Salzmann & Urtasun (2010a), where pairwise products of the original outputs are learned and subsequently transformed back via a singular value decomposition (SVD); for more details on the technicalities we refer to Section C.5 in the Supplementary material.

A visualization of the problem is provided in Figure 2 where different poses $\alpha$ of the triangle are depicted; the blue points represent the corners of the triangle, the positions of which are learned by the GP. As can be seen from the data in Table 2, our approach here performs best for low noise levels. When the noise is very small, $\sigma \lesssim 1e-3$, the constrained approach achieves about the same overall accuracy in terms of RMSE as the unconstrained GP, whereas the error in the constraint is reduced by factors of 2-6.

This reduction is not simply a result of the particular parameterization of the problem, which enforces the constraint implicitly for noiseless observations, as shown by Salzmann & Urtasun (2010a). To see that, we included the results for a GP that is trained on the transformed outputs, but where the constraint is not enforced explicitly. Table 2 shows that the result is improved when enforcing the constraint in addition to using the transformed outputs.

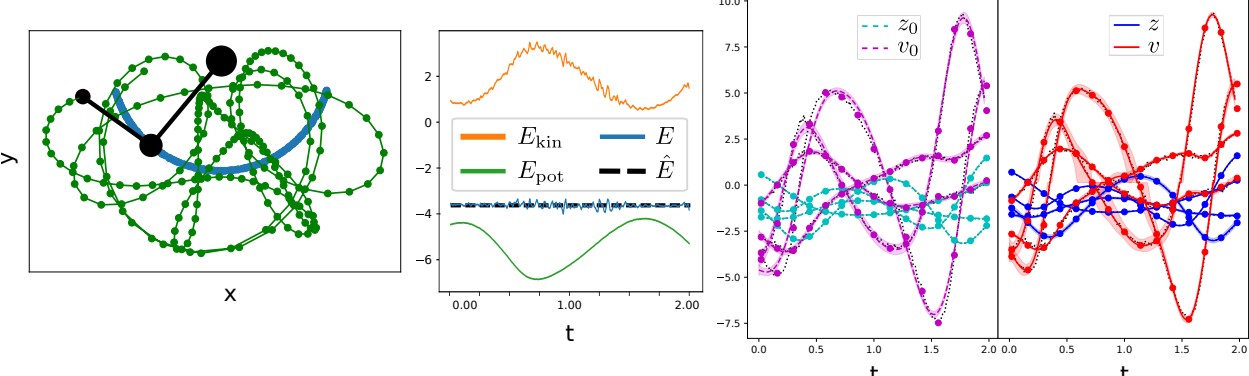

Figure 3: **Left:** Trajectory of the double pendulum. Note that the trajectory shown here is longer than the sequences of motion considered in the plots to the right. **Middle:** Kinetic energy $E_{\mathrm{kin}}$, potential energy $E_{\mathrm{pot}}$ and total energy $E$ of the double pendulum are shown. It is apparent that for the considered segment of the motion the energy is constant for practical purposes, except for fluctuations in the contribution of the kinetic energy due to measurement error. An estimate $\hat{E}$ of the energy is obtained by averaging over $E$. **Right:** Positions $z_0$, $z$ and velocities $v_0$, $v$ of the masses (four components each) as learned by the unconstrained (left inset) and the constrained GP (right inset), respectively; the posterior means of the GPs are depicted together with the $2\sigma$ credible intervals. The dotted lines represent the available data, where the subset of big dots has been used for training.

## 4.3   Real Data Experiment: Double Pendulum

In this section we consider the 'Double Pendulum Chaotic' dataset (Asseman et al., 2018); this dataset consists of 21 different two dimensional trajectories of a double pendulum and contains annotated positions of the masses attached at the ends of the two pendula. Each trajectory consists of about 17000 measurements, taken at a frequency of 500 Hz. For more information on the parameters of the double pendulum, see Section D.1 in the Supplementary material. We attempt to construct a GP that models both positions $z_x$, $z_y$ and velocities $v_x$, $v_y$ of the two masses (i.e. 8 outputs), while at the same time respecting the law of energy conservation; the time $t$ serves as input. As friction is present, we consider a limited section of the trajectory during the second half of the motion where we can assume constant energy (compare Figure 3); energy conservation here takes the form

$$E = m_b g z_{by} + m_g g z_{gy} + \frac{m_b}{2}\left(v_{bx}^2 + v_{by}^2\right) + \frac{m_g}{2}\left(v_{gx}^2 + v_{gy}^2\right), \tag{11}$$

where $g$ denotes the gravitational acceleration on earth, and where the indices $b$ and $g$ refer to the blue and the green pendulum, respectively. The constraint is incorporated into the GP in analogy to the harmonic oscillator. In terms of (6), we identify $a_1 = 0$, $a_2 = m_b g$, $a_3 = 0$, $a_4 = m_g g$, $a_5 = m_b/2$, $a_6 = m_b/2$, $a_7 = m_g/2$, $a_8 = m_g/2$, $h_2(z_{by}) = z_{by}$, $h_4(z_{gy}) = z_{gy}$, $h_5(v_{bx}) = v_{bx}^2$, $h_6(v_{by}) = v_{by}^2$, $h_7(v_{gx}) = v_{gx}^2$, $h_8(v_{gy}) = v_{gy}^2$ and $C = E$; note that the coefficients $a_1$, $a_3$ correspond to the outputs $z_{bx}$, $z_{gx}$, which are not part of the constraint (11).

We pick a sequence of 200 data points (which are fairly close together) from one of the trajectories; 15 of these points are used during hyperparameter optimization, and to receive an estimate $\hat{E}$ of the energy. The remaining 185 points are used as test data to compare the performance of constrained and unconstrained GP, both in terms of constraint fulfillment and in terms of RMSE with respect to the data.

Results for one individual sequence are shown in the rightmost plot of Figure 3. We observe that the constrained GP is better at learning the precise shapes of the extrema of the velocity curves, although some artefacts arise close to zero crossings due to inaccurately learned square values. For values close to zero, the credible intervals of the unconstrained GP are often smoother and thinner than those of the constrained GP.

Averaging the results over 50 sequences chosen at random from the second half of the trajectories (with less friction), the RMSE for the constrained GP is $0.31 \pm 0.14$, whereas for the unconstrained GP it is $0.33 \pm 0.16$.

In terms of constraint fulfillment, the constrained GP clearly performs better with $|\Delta C| = 0.17 \pm 0.14$ as compared to $|\Delta C| = 0.91 \pm 0.61$ for the unconstrained GP. The values here are given plus-or-minus one standard deviation.

## 5 Related Work

Several research projects have considered incorporating constraints into the GP; examples include boundary conditions (Solin & Kok, 2019), inequality constraints (Veiga & A.Marrel, 2012; Maatouk & Bay, 2017) and differential equation constraints (Jidling et al., 2017; Raissi et al., 2017; 2018). The recent review by Swiler et al. (2020) provides a good overview of the existing literature on constrained GPs. So far, most of the efforts have been concentrated on the single-task GP. The sum constraint, however, is qualitatively very different from constraints on single-task GPs, in that it explicitly enforces a relationship between different outputs instead of acting on individual outputs. Hence, in this section, we focus on works that consider constraints on the outputs of multitask GPs.

Prior knowledge about vector fields have been imposed into GPs through special divergence-free and curl-free kernels (Wahlström et al., 2013). Jidling et al. (2017) developed a more general method to include linear operator constraints into the kernel of the GP; this is possible by using the property that GPs are closed under linear transformations (Papoulis & Pillai, 2001) and relating the GP to a suitable latent GP, resembling the use of potential functions in physics. See also Lange-Hegermann (2018) for a discussion of this approach from a more mathematical perspective. Practical applications include modelling of electromagnetic fields (Solin et al., 2018) and reconstruction of strain fields (Jidling et al., 2018; Hendriks et al., 2019b;a; 2020b). Geist & Trimpe (2020) consider affine constraints on the dynamics of mechanical systems and construct a GP satisfying Gauss' principle of least constraint.

There is a connection between our method and the method by Jidling et al. (2017): while they do not consider affine constraints, their approach can be extended to include those in the context of the constant linear sum constraint (compare also Hendriks et al. (2020a), where the same idea is applied to neural networks). The two works attack the problem from different angles: whereas Jidling et al. (2017) start by directly constructing a covariance matrix out of vectors spanning the nullspace of the constraining operator, we start with the covariance matrix and subsequently constrain it. More details on these parallels are given in Appendix E. An advantage of our approach is that it is straightforward to include additional structure in the task kernel, such as in (B.12). Furthermore, we consider the general case of nonconstant, nonlinear sum constraints.

Constructing kernels that are invariant with respect to certain symmetry transformations has proven fruitful in the fields of atomic and molecular physics. Glielmo et al. (2017) consider GPs to model interatomic force fields; they construct a 'covariant kernel' by including symmetries of the force, such as rotation and reflection. Methods for constructing invariant kernels are given by Haasdonk & Burkhardt (2007), whereas Chmiela et al. (2020) use a similar approach to construct a kernel that allows for simultaneous prediction of energies and forces in molecules.

Pose estimation constitutes another area where constrained multitask GPs are of importance; in the case of rigid pose estimation, the lengths are required to be constant. A method to explicitly enforce the constraints during inference is given by Salzmann & Urtasun (2010b), whereas Salzmann & Urtasun (2010a) propose a method to implicitly enforce the fixed-length constraint by learning transformed outputs in which the constraint is linear. We followed this latter approach in the example with the rotated triangle in Section 4.2; in addition to using the transformed outputs we also imposed the length constraint explicitly, which (at least in principle) should allow for training points that do not fulfill the constraint exactly.

## 6 Conclusions and Future Work

We have derived a way of incorporating both linear and nonlinear sum constraints into multitask GPs. This is achieved by learning transformed outputs and by conditioning the prior distribution of the GP on the constraint. The toy problem of the harmonic oscillator demonstrated the potential of the method; it showed that the constraint is fulfilled with high accuracy and that the constrained GP can mitigate detrimental

effects of noise or of incomplete measurements. Our experiment with the triangle in the plane showed that the sum constraint improved the method by Salzmann & Urtasun (2010a) of including the length constraint into pose estimation problems; so far, these results are particularly promising in the low-noise setting. The results for the double pendulum dataset showed that our method also works well in case of real-world, noisy data, given a way of estimating the constraint with sufficient accuracy.

In light of the results received for the triangle in the plane in Section 4.2, it appears as if it would be worth investigating the applicability of this approach to pose estimation problems further; especially, in cases where the approach by Salzmann & Urtasun (2010a) gives good results, our constrained GP could potentially improve the performance. To increase the suitability of the approach for big datasets, combining the sum constraint framework with methods such as sparse variational inference (Hensman et al., 2013) appears to be a fruitful direction of inquiry. Finding general methods to incorporate constraints similar to the length constraint (10) into the GP, where nonlinearities may depend on more than one of the outputs at once, constitutes another interesting avenue of future research and would widen the range of possible applications.

## Acknowledgements

The work is financially supported by the Swedish Research Council (VR) via the project *Physics-informed machine learning* (registration number: 2021-04321) and by the *Kjell och Märta Beijer Foundation.*

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

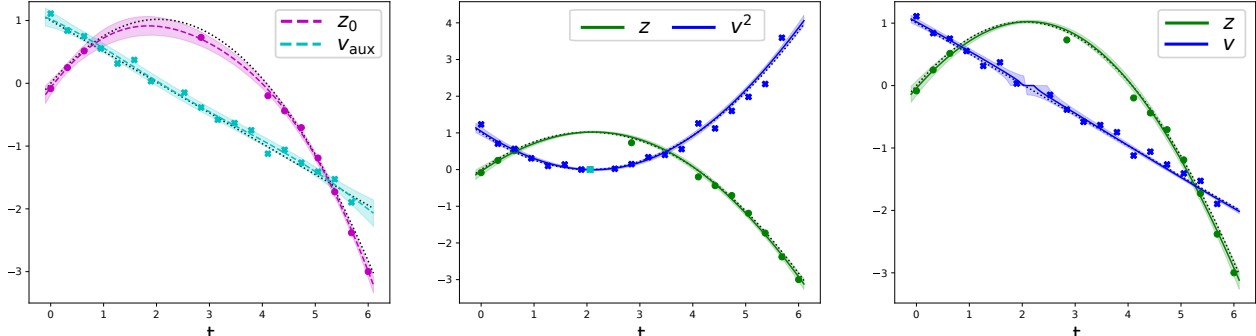

Figure 4: Demonstration of the auxiliary variables approach for the free fall. The quantities $z_0$, $z$ and $v_{\text{aux}}$, $v$ refer to the position and velocity of the mass, respectively. We distinguish between $z_0$, $z$ and $v_{\text{aux}}$, $v$ to emphasize that, while they aim to approximate the same curve, they are learned by different GPs. The posterior means of the GPs are depicted, together with the 2-sigma credible intervals. The dotted lines represent the true curves and the big dots/crosses correspond to the data available to the GPs. **Left:** Results for the unconstrained GP are shown. For this example, the $v$-curve coincides with the auxiliary variable $v_{\text{aux}}$ required for the constrained GP. **Middle:** The transformed outputs learned by the constrained GP are depicted. The results for the auxiliary variable have been employed to create a virtual measurement for $v^2$ at the zero crossing of $v_{\text{aux}}$ (differently colored square) in order to force the quadratic function towards zero. **Right:** The backtransformed outputs of the constrained GP are shown, where the auxiliary output $v_{\text{aux}}$ has been used to recover the sign of $v$.

|  |  | $\sigma_n = 0.05$ | | $\sigma_n = 0.1$ | | $\sigma_n = 0.3$ | | |
|---|---|---|---|---|---|---|---|---|
|  |  | GP-c | GP-u | GP-c | GP-u | GP-c | GP-u | |
| $f_d = 0$ | RMSE | **1.9**±0.5 | 2.3±0.4 | **3.2**±1.1 | 4.7±1.2 | **10.1**±3.6 | 13.2±3.3 | (e-2) |
|  | $\|\Delta C\|$ | **0.7**±2.6 | 35.3±13.6 | **0.1**±0.1 | 79.0±31.8 | **0.0**±0.1 | 216.1±84.0 | (e-2) |
| $f_d = 0.3$ | RMSE | **2.5**±0.9 | 3.4±1.2 | **3.9**±1.6 | 6.0±2.0 | **15.3**±7.9 | 20.0±7.8 | (e-2) |
|  | $\|\Delta C\|$ | **0.4**±0.8 | 47.5±26.8 | **0.1**±0.1 | 93.7±35.1 | **0.7**±2.1 | 286.9±123.8 | (e-2) |

Table 3: Comparison of the performance of the constrained GP (GP-c) and the unconstrained GP (GP-u) for the free fall. Shown are the root mean squared error (RMSE) of the prediction as well as the mean absolute violation of the constraint, $|\Delta C|$. The standard deviation of the noise is given by $\sigma_n$ whereas $f_d$ is the probability with which output components have been omitted at random from the data. The values have been obtained by averaging over 50 datasets and are given plus-or-minus one standard deviation. Bold font highlights best performance.

## A   Additional Examples

In this Section, we take a look at some additional examples where the sum constraint can be applied. The free fall dataset in Section A.1 is another example from physics, where one of the outputs enters linearly into the constraint, instead of quadratically. The damped harmonic oscillator in Section A.2 constitutes a variation of the harmonic oscillator toy example and demonstrates the case of a non-constant constraint. In Section A.3, we investigate an example where the constraint includes different nonlinearities.

### A.1   Free Fall

In addition to the harmonic oscillator (see Section 4.1), we investigated the simple example of a mass in free fall as a second toy problem. Here, the output of the GP consists in position and velocity of the mass,

$\mathbf{f}^\mathsf{T} = [z, v]$, whereas the time $t$ serves as input. Then the constraint takes the following form

$$\mathcal{F}[\mathbf{f}(\mathbf{t})] = mgz(t) + \frac{m}{2}v(t)^2 = E_{\text{pot}}(t) + E_{\text{kin}}(t) = E. \tag{A.1}$$

In terms of (6), we identify $a_1 = mg$, $a_2 = m/2$, $h_1(z) = z$, $h_2(v) = v^2$ and $C = E$. Hence, we receive for the transformed outputs $f_1' = f_1 = z$ and $f_2' = v^2$. We choose the auxiliary output as $f_{\text{aux}}^1 = v$, which we use to extract the sign when backtransforming $f_2'$ and to create virtual measurements for $f_2'$ at the zero crossings of the posterior mean of $f_{\text{aux}}^1$. In order to fit the transformed outputs of the GP, the observations $\mathbf{y}_k = [z_k, v_k]^\mathsf{T}$ are transformed analogously to obtain $\mathbf{y}_k' = [z_k, v_k^2, v_k]^\mathsf{T}$; $v_k$ is part of the constrained outputs, as this improves the performance for this example. The virtual measurements are also included in the transformed data $\mathbf{y}'$. In terms of the transformed outputs the constraint can be written compactly as $\mathbf{F}\mathbf{f}' = C$, where $\mathbf{F} = [a_1, a_2, 0]$. For more details on the free fall dataset, see Section C.3.

In Figure 4, results for both constrained and unconstrained GP, applied to the free fall dataset, are depicted. When comparing the left and the right plot, it is apparent, that the constrained GP manages to mitigate detrimental effects of both noise and incomplete measurements, where some of the observed output components have been omitted at random, better than the unconstrained GP (compare area around peak of $z_0$ in the figure). When looking at the $2\sigma$ credible intervals, we get the same picture as before for the harmonic oscillator: the constrained GP can utilize higher certainty in one output and transfer it to the other one, resulting in overall slimmer intervals. Close to the zero crossing of $v$, however, some artefacts are present due to the piecewise, nonlinear backtransformation, which are absent for the unconstrained GP; furthermore, confidence intervals are stretched a bit due to the backtransformation via the square root.

In Table 3, values for both the root mean squared error (RMSE) and the average absolute violation of the constraint $|\Delta C|$ are given for various noise levels $\sigma_n$, both with complete and incomplete measurements; in case of incomplete measurements, the output components have been omitted at random with probability $f_d = 0.3$. The values have been obtained by averaging over 50 datasets. We observe that the constrained GP fulfills the constraint with up to two orders of magnitude higher accuracy and also performs better in terms of RMSE.

## A.2 Damped Harmonic Oscillator

Next, we investigate a slight variation of the harmonic oscillator, the damped harmonic oscillator. The formal treatment remains mostly unchanged and details can be found in Section 3.1.2 in the main paper; the main difference is that damping has been added to the model of the oscillator. As a consequence, the energy is no longer constant and the amplitude of the oscillation decays over time; see Section C.2 for more details. Hence, this example constitutes an instance of the non-constant sum constraint $\mathcal{F}[\mathbf{f}(\mathbf{t})] = E(t)$, where Algorithm 2 applies.

In Figure 5, results for both constrained and unconstrained GP are depicted. The findings are similar to the undamped harmonic oscillator, and it is apparent that the constrained GP can mitigate the detrimental effects of noisy or incomplete measurements better than the unconstrained GP. In Table 4, the performance on 50 random datasets is evaluated. The outputs of the constrained GP fulfill the constraint with up to two orders of magnitude higher accuracy than the unconstrained one, and also perform slightly better in terms of RMSE. This example demonstrates that, given similar datasets, the performance of our method is very similar, both in case of constant and non-constant constraints (compare Section 4.1).

## A.3 Non-square Nonlinearity

Finally, we investigate an example where nonlinearities other than the square-nonlinearity are involved in the constraint. We consider the outputs $\mathbf{f} = \begin{bmatrix} f_1, f_2 \end{bmatrix}^\mathsf{T}$, on which we want to enforce the constraint

$$\mathcal{F}[\mathbf{f}(\mathbf{x})] = \log(f_1(x)) + \sin(f_2(x)) = C(x). \tag{A.2}$$

In terms of (6), we identify $a_1 = 1$, $a_2 = 1$, $h_1(f_1) = \log(f_1)$, $h_2(f_2) = \sin(f_2)$ and $C = C(x)$. Here, we assume that the true value $C(x)$ is known. Hence, we receive for the transformed outputs $f_1' = \log(f_1)$ and $f_2' = \sin(f_2)$.

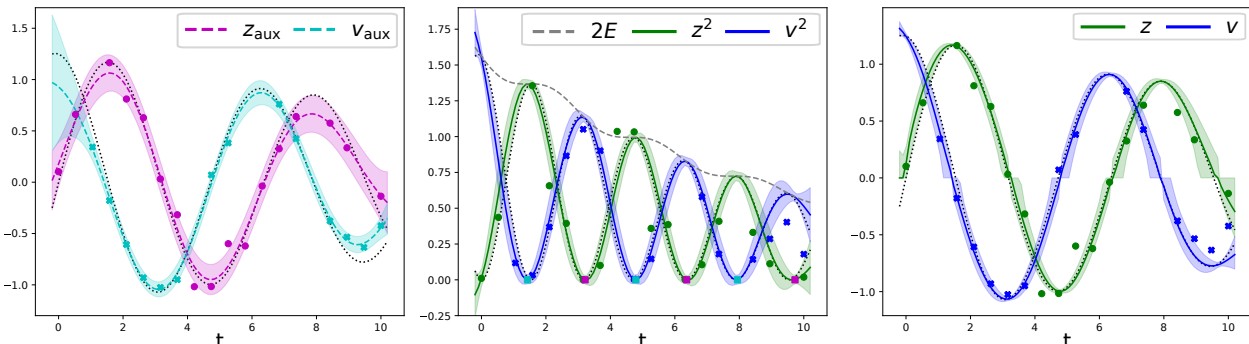

Figure 5: Demonstration of the auxiliary variables approach for the damped harmonic oscillator. The quantities $z_{\text{aux}}$, $z$ and $v_{\text{aux}}$, $v$ refer to the position and velocity of the damped harmonic oscillator, respectively. We distinguish between $z_{\text{aux}}$, $z$ and $v_{\text{aux}}$, $v$ to emphasize that, while they aim to approximate the same curve, they are learned by different GPs. The posterior means of the GPs are depicted, together with the $2\sigma$ credible intervals. The dotted lines represent the true curves and the big dots/crosses correspond to the data available to the GPs. **Left:** Results for the unconstrained GP are shown. For this example, these outputs coincide with the auxiliary outputs required for the constrained GP. **Middle:** The transformed outputs learned by the constrained GP are depicted, together with the constraint $2E(t) = kz^2 + mv^2$ (where $k = 1$ and $m = 1$). The results for the auxiliary outputs have been employed to create virtual measurements at zero crossings (differently colored squares) in order to force the quadratic functions towards zero. **Right:** The backtransformed outputs of the constrained GP are shown, where the auxiliary outputs $z_{\text{aux}}$ and $v_{\text{aux}}$ have been used to recover the signs.

|  |  | $\sigma_n = 0.05$ | | $\sigma_n = 0.1$ | | $\sigma_n = 0.3$ | | |
|---|---|---|---|---|---|---|---|---|
|  |  | GP-c | GP-u | GP-c | GP-u | GP-c | GP-u | |
| $f_d = 0$ | RMSE | **3.1**±1.3 | 3.2±0.6 | **5.6**±2.3 | 6.5±1.2 | **13.4**±4.1 | 17.7±3.8 | (e-2) |
|  | $\|\Delta C\|$ | **0.0**±0.0 | 2.4±0.6 | **0.0**±0.0 | 5.1±1.1 | **0.1**±0.1 | 13.3±3.3 | (e-2) |
| $f_d = 0.2$ | RMSE | **4.1**±2.3 | 4.8±2.2 | **6.2**±2.6 | 7.8±2.1 | **20.7**±11.4 | 23.5±7.1 | (e-2) |
|  | $\|\Delta C\|$ | **0.1**±0.1 | 3.3±0.9 | **0.1**±0.1 | 5.8±1.4 | **0.2**±0.3 | 16.3±4.8 | (e-2) |

Table 4: Comparison of the performance of the constrained GP (GP-c) and the unconstrained GP (GP-u) for the damped harmonic oscillator. Shown are the root mean squared error (RMSE) of the prediction as well as the mean absolute violation of the constraint, $|\Delta C|$. The standard deviation of the noise is given by $\sigma_n$ whereas $f_d$ is the probability with which output components have been omitted at random from the data. The values have been obtained by averaging over 50 datasets and are given plus-or-minus one standard deviation. Bold font highlights best performance.

We choose the auxiliary output as $f^1_{\text{aux}} = f_2$, which we use to disambiguate the backtransformation via the arcsine, that is we keep track of how many multiples of $\pm\pi/2$ the output $f^1_{\text{aux}}$ has crossed. We also use the auxiliary output to create virtual measurements for $f'_2$ at points where the posterior mean of $f^1_{\text{aux}}$ crosses multiples of $\pm\pi/2$, in order to reduce artefacts caused by the discontinuity in the backtransformation.

In Figure 6, results for both constrained and unconstrained GP are depicted. It is apparent that, while not perfect, the constrained GP outperforms the unconstrained one. In Table 5, the results averaged over 50 datasets are given. We see, that the constrained GP outperforms the unconstrained one in terms of RMSE, and it fulfills the constraint with up to 30 times higher accuracy.

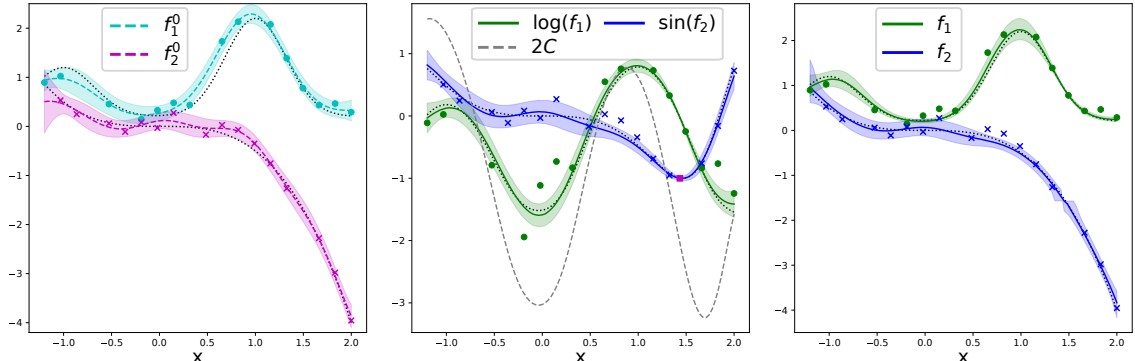

Figure 6: Demonstration of the auxiliary variables approach for the example with non-square nonlinearity. The quantities $f_1^0$, $f_1$ and $f_2^0$, $f_2$ refer to the same respective outputs of the GPs. We distinguish between $f_1^0$, $f_1$ and $f_2^0$, $f_2$ to emphasize that, while they aim to approximate the same curve, they are learned by different GPs. The posterior means of the GPs are depicted, together with the 2-sigma credible intervals. The dotted lines represent the true curves and the big dots/crosses correspond to the data available to the GPs. **Left:** Results for the unconstrained GP are shown. For this example, the $f_2^0$-curve coincides with the auxiliary output $f_{\text{aux}}$ required for the constrained GP. **Middle:** The transformed outputs learned by the constrained GP are depicted, together with the constraint $C = \log(f_1) + \sin(f_2)$. The result for the auxiliary output has been employed to create a virtual measurement at the point where $f_{\text{aux}}$ crosses $-\pi/2$ (differently colored square). **Right:** The backtransformed outputs of the constrained GP are shown, where the auxiliary output $f_{\text{aux}}$ has been used to disambiguate the backtransformation via the arcsine.

|  |  | $\sigma_n = 0.05$ | | $\sigma_n = 0.1$ | | $\sigma_n = 0.15$ | | |
|---|---|---|---|---|---|---|---|---|
|  |  | GP-c | GP-u | GP-c | GP-u | GP-c | GP-u | |
| $f_d = 0$ | RMSE | **3.0**±1.7 | 3.5±0.5 | **4.6**±1.0 | 7.3±1.1 | **7.0**±1.0 | 10.8±2.2 | (e-2) |
|  | $|\Delta C|$ | **1.3**±2.3 | 5.9±1.4 | **0.9**±0.5 | 13.9±4.8 | **0.5**±0.5 | 18.0±6.3 | (e-2) |
| $f_d = 0.2$ | RMSE | **3.1**±1.0 | 8.6±9.9 | **5.6**±1.8 | 11.7±8.8 | **8.4**±2.1 | 16.2±7.3 | (e-2) |
|  | $|\Delta C|$ | **1.1**±0.8 | 12.2±14.2 | **0.7**±0.6 | 17.2±11.4 | **0.4**±0.4 | 23.5±10.8 | (e-2) |

Table 5: Comparison of the performance of the constrained GP (GP-c) and the unconstrained GP (GP-u) for the example with non-square nonlinearity. Shown are the root mean squared error (RMSE) of the prediction as well as the mean absolute violation of the constraint, $|\Delta C|$. The standard deviation of the noise is given by $\sigma_n$ whereas $f_d$ is the probability with which output components have been omitted at random from the data. The values have been obtained by averaging over 50 datasets and are given plus-or-minus one standard deviation. Bold font highlights best performance.

### A.4 Comparison of approximation methods

In this section, we give a brief comparison of different methods of approximate inference at the example of the harmonic oscillator. The approximation methods under consideration are the Laplace approximation B.3 and variational inference B.4.

Fig. 7 shows the predictive performance of the the unconstrained GP, variational inference and the Laplace approximation. While it is apparent that both approximate GPs fulfill the constraint with high precision, the variational approach tends to overfit to the data. On the other hand, overconfident credible intervals seem to be less of an issue for the variational approach than for the Laplace approximation.

In Table 6, results obtained when averaging over 20 runs are given for different noise settings. It is apparent that the constrained GP with Laplace approximation performs best. While the constrained GP utilizing variational inference performs worst in terms of root-mean-square error, the constraint is still fulfilled with high precision. In case of the variational approach, it might be possible to improve upon these results by

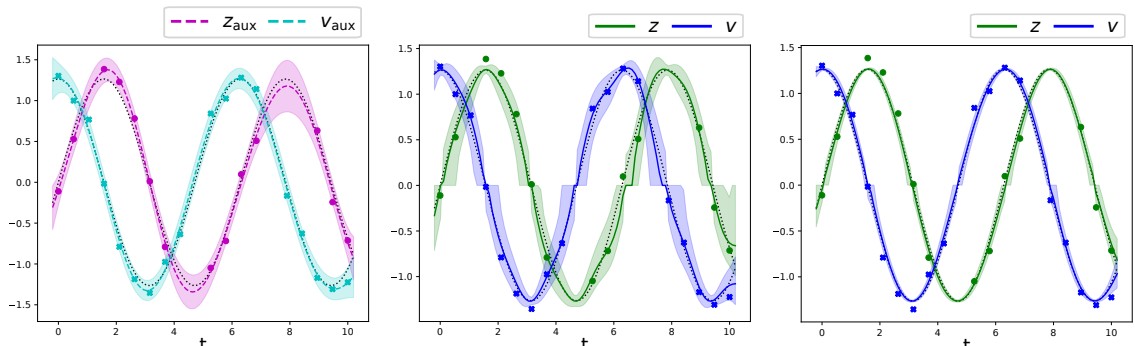

Figure 7: Comparison of the performance of the unconstrained GP (**Left**), the variational approach (**Middle**), and the Laplace approximation (**Right**) for the example of the harmonic oscillator. The quantities $z_{\mathrm{aux}}$, $z$ and $v_{\mathrm{aux}}$, $v$ refer to the position and velocity of the harmonic oscillator, respectively. The posterior means of the GPs are depicted, together with the $2\sigma$ credible intervals. The dotted lines represent the true curves and the big dots/crosses correspond to the data available to the GPs.

|  |  | $\sigma_n = 0.05$ | | | $\sigma_n = 0.1$ | | | |
|---|---|---|---|---|---|---|---|---|
|  |  | GP-c L | GP-c var | GP-u | GP-c L | GP-c var | GP-u | |
| $f_d = 0$ | RMSE | **2.4**±0.7 | 3.8±0.6 | 3.2±0.4 | **4.2**±1.1 | 7.0±1.6 | 6.3±0.9 | (e-2) |
| | $\|\Delta C\|$ | **0.0**±0.0 | **0.0**±0.0 | 3.3±0.7 | **0.0**±0.0 | 0.1±0.1 | 6.4±1.4 | (e-2) |
| $f_d = 0.2$ | RMSE | **3.0**±1.2 | 5.5±3.0 | 4.5±1.6 | **6.7**±4.0 | 9.1±3.1 | 8.3±2.2 | (e-2) |
| | $\|\Delta C\|$ | **0.0**±0.0 | 0.2±0.3 | 4.2±0.9 | **0.1**±0.2 | 0.4±0.6 | 8.1±1.7 | (e-2) |

Table 6: Comparison of the performance of the constrained GP with Laplace approximation (GP-c L), the constrained GP using variational inference (GP-c var), and the unconstrained GP (GP-u) for the harmonic oscillator. Shown are the root mean squared error (RMSE) of the prediction as well as the mean absolute violation of the constraint, $|\Delta C|$. The standard deviation of the noise is given by $\sigma_n$ whereas $f_d$ is the probability with which output components have been omitted at random from the data. The values have been obtained by averaging over 20 datasets and are given plus-or-minus one standard deviation. Bold font highlights best performance.

trying different parameterizations of the variational distribution, or by finding a better suited optimization scheme.

# B Technicalities

## B.1 Background on GP Regression

In this section we give a very brief overview of some important GP regression formulas. For more detailed accounts see e.g. Rasmussen & Williams (2006); Lindholm et al. (2021). Given the mean function $m(\cdot)$ and kernel $K(\cdot, \cdot)$ of the GP, the predictive distribution of the GP can be calculated by first constructing the joint distribution between observations $\mathbf{y}$ and function values at test locations $\mathbf{f}_*$,

$$\begin{bmatrix} \mathbf{y} \\ \mathbf{f}_* \end{bmatrix} \sim \mathcal{N} \left( \begin{bmatrix} \mathbf{m} \\ \mathbf{m}_* \end{bmatrix}, \begin{bmatrix} \mathbf{K} + \sigma_n^2 \mathbf{I} & \mathbf{K}_* \\ \mathbf{K}_*^\mathsf{T} & \mathbf{K}_{**} \end{bmatrix} \right), \tag{B.1}$$

where $\mathbf{m} = m(\mathbf{X})$, $\mathbf{m}_* = m(\mathbf{X}_*)$, $\mathbf{K} = K(\mathbf{X}, \mathbf{X})$, $\mathbf{K}_* = K(\mathbf{X}, \mathbf{X}_*)$ and $\mathbf{K}_{**} = K(\mathbf{X}_*, \mathbf{X}_*)$.

Then, the conditional distribution $\mathbf{f}_*|\mathbf{X}, \mathbf{y}, \mathbf{X}_*$ is constructed as follows:

$$\mathbf{f}_*|\mathbf{X}, \mathbf{y}, \mathbf{X}_* \sim \mathcal{N}\left(\bar{\mathbf{f}}_*, \mathrm{cov}(\mathbf{f}_*)\right), \text{ where} \tag{B.2a}$$

$$\bar{\mathbf{f}}_* \triangleq \mathbb{E}[\mathbf{f}_*|\mathbf{X}, \mathbf{y}, \mathbf{X}_*]$$

$$= \mathbf{m}_* + \mathbf{K}_*^\mathsf{T}[\mathbf{K} + \sigma_n^2\mathbf{I}]^{-1}(\mathbf{y} - \mathbf{m}), \tag{B.2b}$$

$$\mathrm{cov}(\mathbf{f}_*) = \mathbf{K}_{**} - \mathbf{K}_*^\mathsf{T}[\mathbf{K} + \sigma_n^2\mathbf{I}]^{-1}\mathbf{K}_*. \tag{B.2c}$$

The log-marginal likelihood, which is used for hyperparameter optimization, is given by

$$\log p(\mathbf{y}|\mathbf{X}) = -\frac{1}{2}(\mathbf{y} - \mathbf{m})^\mathsf{T}(\mathbf{K} + \sigma_n^2\mathbf{I})^{-1}(\mathbf{y} - \mathbf{m})$$

$$-\frac{1}{2}\log|\mathbf{K} + \sigma_n^2\mathbf{I}| - \frac{n}{2}\log 2\pi. \tag{B.3}$$

## B.2 Accommodating incomplete measurements

Throughout the paper, we often consider the case of incomplete measurements, i.e. data points where measurements are available only for a subset of the tasks. This can be taken into account by considering equation (B.1) and removing the the rows and columns on the right-hand side corresponding to missing entries in $\mathbf{y}$.

To make this more concrete, let us assume that the j-th entry of $\mathbf{y}$ is missing on the left-hand side of (B.1). Then we also delete the j-th row of $\mathbf{m}$, $(\mathbf{K} + \sigma_n^2\mathbf{I})$ and $\mathbf{K}_*$, as well as the j-th column of $(\mathbf{K} + \sigma_n^2\mathbf{I})$ and $\mathbf{K}_*^\mathsf{T}$, before explicitly constructing the joint distribution. We proceed analogously when training the constrained GP on the transformed data $\mathbf{y}'$.

## B.3 Laplace approximation

The Laplace approximation can be employed when the noise distribution corresponding to the (transformed) observations $\mathbf{y}'$ is non-Gaussian in order to obtain analytical expressions for the predictive equations and for the log-marginal likelihood. Following again Rasmussen & Williams (2006), we approximate the posterior $p(\mathbf{f}'|\mathbf{y}') \propto p(\mathbf{y}'|\mathbf{f}')p(\mathbf{f}')$ via $p(\mathbf{f}'|\mathbf{y}') \approx q(\mathbf{f}'|\mathbf{y}') = \mathcal{N}\left(\mathbf{f}'|\hat{\mathbf{f}}', (\mathbf{K}^{-1} + \mathbf{W})^{-1}\right)$, where

$$\hat{\mathbf{f}}' = \mathbf{K}\left(\nabla_{\mathbf{f}'}\log p(\mathbf{y}'|\mathbf{f}')\right)|_{\mathbf{f}'=\hat{\mathbf{f}}'}, \tag{B.4a}$$

$$\mathbf{W} = -\nabla_{\mathbf{f}'}\nabla_{\mathbf{f}'}\log p(\mathbf{y}'|\mathbf{f}')|_{\mathbf{f}'=\hat{\mathbf{f}}'}. \tag{B.4b}$$

Newton's method is employed to iteratively determine $\hat{\mathbf{f}}'$ from (B.4a) via the update rule

$$\mathbf{f}'^{\mathrm{new}} = \mathbf{f}' - \gamma\left(\nabla_{\mathbf{f}'}\nabla_{\mathbf{f}'}\Psi(\mathbf{f}')\right)^{-1}\nabla_{\mathbf{f}'}\Psi(\mathbf{f}') \tag{B.5}$$

$$= \gamma\mathbf{m} + (1-\gamma)\mathbf{f}' + \gamma\left((\mathbf{K}^{-1} + \mathbf{W})^{-1}(\nabla_{\mathbf{f}'}\log p(\mathbf{y}'|\mathbf{f}') + \mathbf{W}(\mathbf{f}' - \mathbf{m}))\right), \tag{B.6}$$

where $\Psi(\mathbf{f}') = \log p(\mathbf{y}'|\mathbf{f}') + \log p(\mathbf{f}'|\mathbf{X})$ and where $\gamma$ is the step size. In terms of these quantities, the expressions (B.2) from the previous section become

$$\bar{\mathbf{f}}_*' = \mathbf{m} + \mathbf{K}_*^\mathsf{T}\mathbf{K}^{-1}(\hat{\mathbf{f}}' - \mathbf{m}), \tag{B.7a}$$

$$\mathrm{cov}(\mathbf{f}_*') = \mathbf{K}_{**} - \mathbf{K}_*^\mathsf{T}[\mathbf{K} + \mathbf{W}^{-1}]^{-1}\mathbf{K}_*, \tag{B.7b}$$

$$\log p(\mathbf{y}'|\mathbf{X}) = -\frac{1}{2}(\hat{\mathbf{f}}' - \mathbf{m})^\mathsf{T}\mathbf{K}^{-1}(\hat{\mathbf{f}}' - \mathbf{m}) + \log p(\mathbf{y}'|\hat{\mathbf{f}}') - \frac{1}{2}\log(|\mathbf{K}||\mathbf{K}^{-1} + \mathbf{W}|). \tag{B.7c}$$

For details on the derivation of these formulas, see Section 3.4 in Rasmussen & Williams (2006).

For the Laplace approximation, the likelihood $p_{\mathbf{y}'}(\mathbf{y}'|\mathbf{f}')$ of the transformed data $\mathbf{y}' = h(\mathbf{y})$ has to be known. We assume the original data $\mathbf{y}$ to be contaminated by Gaussian noise. In case of the square nonlinearity

where $\mathbf{y}' = \mathbf{y}^2$, the likelihood is then given by the pdf of a noncentral chi-squared distribution. In the case of arbitray nonlinearities $h_j$, the likelihood can be obtained via

$$p_{\mathbf{y}'}(\mathbf{y}'|\mathbf{f}') = \prod_{ij} p_{y'_{ij}}(y'_{ij}|f'_{ij}) = \prod_{ij} p_{y_{ij}}\left(h_j^{-1}(y'_{ij})|h_j^{-1}(f'_{ij})\right)\left|\frac{dh_j^{-1}(y'_{ij})}{dy'_{ij}}\right|, \tag{B.8}$$

where the indices $i$ and $j$ denote data points and tasks, respectively.

There is no guarantee that Newton's method will determine the correct maximum $\hat{\mathbf{f}}'$ in case of multimodal distributions, or that the resulting Gaussian distribution will constitute a good approximation of the true posterior. For these reasons, it has to be decided on a case by case basis whether the Laplace approximation should be employed or not. Visual inspection of the GP predictions often gives a good idea on whether the Laplace approximation performs well or not. In cases where it does not perform well, standard GP regression might still produce reasonable results. Alternatively, different methods such as variational inference (**?**) or expectation propagation (Minka, 2001) could be employed; the equations in (B.7) will then need to be replaced by expressions corresponding to these techniques. A brief discussion on variational inference is given in Appendix B.4, as well as a comparison with the Laplace approximation in Appendix A.4.

Throughout the paper, we used the Laplace approximation for the harmonic oscillator 4.1, the free fall A.1, the damped harmonic oscillator A.2, and the example with non-square nonlinearity A.3. In case of the double pendulum 4.3 and the triangle in the plane 4.2, we chose standard GP regression over the Laplace approximation.

## B.4   Variational Inference

As an alternative to the Laplace approximation (see previous section), variational inference (Blei et al., 2017; Titsias & Lawrence, 2010) can be employed to approximate the posterior $p(\mathbf{f}'|\mathbf{y}') \propto p(\mathbf{y}'|\mathbf{f}')p(\mathbf{f}')$ when the likelihood $p(\mathbf{y}'|\mathbf{f}')$ is non-Gaussian. The idea is to approximate the posterior with the variational distribution $q(\mathbf{f}') \sim \mathcal{N}\left(\mathbf{f}'|\boldsymbol{\mu}_q, \boldsymbol{\Sigma}_q\right)$ and to learn the parameters $\boldsymbol{\mu}_q, \boldsymbol{\Sigma}_q$ by minimizing the Kullback-Leibler divergence $\text{KL}(q(\mathbf{f}')||p(\mathbf{f}'|\mathbf{y}'))$ between variational distribution and posterior. In order to ensure positive definiteness, the entries of the covariance matrix $\boldsymbol{\Sigma_q}$ are not learned directly, but instead the entries of its Cholesky factor $\mathbf{L_q}$, where it holds that $\boldsymbol{\Sigma_q} = \mathbf{L_q}\mathbf{L_q^\mathsf{T}}$. Since an exact minimization of the KL divergence is intractable, the evidence lower bound $\text{ELBO} = \log p(\mathbf{y}') - \text{KL}(q(\mathbf{f}')||p(\mathbf{f}'|\mathbf{y}')) \leq \log p(\mathbf{y}')$ is maximized in its stead.

The predictive equations in terms of the variational parameters are given by

$$\bar{\mathbf{f}}'_* = \mathbf{m} + \mathbf{K}_*^\mathsf{T}\mathbf{K}^{-1}(\boldsymbol{\mu}_q - \mathbf{m}), \tag{B.9a}$$

$$\text{cov}(\mathbf{f}'_*) = \mathbf{K}_{**} + \mathbf{K}_*^\mathsf{T}\mathbf{K}^{-1}\left(\boldsymbol{\Sigma_q}\mathbf{K}^{-1\mathsf{T}} - \mathbf{I}\right)\mathbf{K}_*, \tag{B.9b}$$

and the ELBO can be rewritten in terms of numerically tractable, one-dimensional integrals

$$\text{ELBO} = \mathbb{E}_q[\log(p(\mathbf{y}'|\mathbf{f}'))] - \text{KL}(q(\mathbf{f}')||p(\mathbf{f}')) \tag{B.10}$$

$$= \int \log(p(\mathbf{y}'|\mathbf{f}'))q(\mathbf{f}')d\mathbf{f}' - \text{KL}(q(\mathbf{f}')||p(\mathbf{f}'))$$

$$= \sum_i \int \log(p(y'_i|f'_i))q(f'_i)df'_i - \text{KL}(q(\mathbf{f}')||p(\mathbf{f}')).$$

Equations B.9 are obtained analogously to (B.7), for a derviation of (B.10), see Titsias & Lawrence (2010). The parameters $\boldsymbol{\mu}_q, \boldsymbol{\Sigma}_q$ of the variational distribution and the GP hyperparameters are determined jointly by maximizing the ELBO, which we do by employing gradient descent. Same as for the Laplace approximation, the likelihood for the transformed data $p(\mathbf{y}'|\mathbf{f}')$ is required when calculating the ELBO and can be obtained via (B.8).

In our experiments, the variational approach tended to overfit to the data more than the Laplace approximation. It is possible that a different parameterization of the variational covariance matrix, or a different optimization

---

**Algorithm 3** Constraining the GP - Special Case of Constant Task Interdependencies

---

**Input:** data mean $m_d(\cdot) = 1$; data kernel $k_d(\cdot, \cdot)$; task mean $\boldsymbol{\mu_t}$; task covariance matrix $\boldsymbol{\Sigma_t}$;
     constraints $(\mathbf{F}, \mathbf{S})$; (transformed) data $\mathbf{X}, \mathbf{y}'$; points of prediction $\mathbf{X}_*$
**Output:** constrained predictive distribution $\mathbf{f}'_*|\mathbf{X}, \mathbf{y}', \mathbf{X}_*$
**Note:** During hyperparameter optimization $\mathbf{X}_* = \{\}$ and hence $\mathbf{f}'_* = \{\}$
**Step 1:** Use $\mathbf{F}, \mathbf{S}$ to calculate constrained $\boldsymbol{\mu}'_{\boldsymbol{t}}, \boldsymbol{\Sigma}'_{\boldsymbol{t}}$ according to (8b)
**Step 2:** Construct parameters $\mathbf{m}, \mathbf{K}$ of the (single task) joint prior distribution according to (B.1)
     -omit noise term $\sigma_n^2 \mathbf{I}$
**Step 3:** Use $\boldsymbol{\mu}'_{\boldsymbol{t}}, \boldsymbol{\Sigma}'_{\boldsymbol{t}}, \mathbf{m}, \mathbf{K}$ to construct constrained (multi task) $\boldsymbol{\mu}'_{\text{tot}}, \boldsymbol{\Sigma}'_{\text{tot}}$ according to (4)
**Step 4:** Remove entries in $\boldsymbol{\mu}'_{\text{tot}}, \boldsymbol{\Sigma}'_{\text{tot}}$ corresponding to incomplete measurements as detailed in Section B.2
**if** *Hyperparameter optimization* **then**
    **Step 5:** Calculate the log marginal likelihood according to (B.7c)
    **Step 6:** Perform optimization step
**else if** *Prediction* **then**
    **Step 5:** Calculate the predictive distribution $\mathbf{f}'_*|\mathbf{X}, \mathbf{y}', \mathbf{X}_*$ according to (B.7a)
**end if**

---

scheme would manage to yield better results. In the paper, we went with the Laplace approximation over variational inference; for a brief comparison of the two approaches at the example of the harmonic oscillator, see Appendix A.4.

## B.5   Kernel and Mean

Throughout the paper we use a radial basis function (RBF) kernel (also: squared exponential kernel) as data kernel,

$$k_{\text{RBF}}(\mathbf{x}, \mathbf{x}') = \sigma_f^2 \exp\left(-\frac{||\mathbf{x} - \mathbf{x}'||^2}{2l^2}\right), \tag{B.11}$$

where $\sigma_f$ is a scale factor and $l$ is the length scale. We use the position independent index kernel provided by gpytorch (Gardner et al., 2018),

$$\mathbf{k_t} = \mathbf{BB}^{\mathsf{T}} + \text{diag}(\mathbf{v}), \tag{B.12}$$

where $\mathbf{B}$ is a low-rank matrix and $\mathbf{v}$ is a non-negative vector; we chose the rank of $\mathbf{B}$ to be equal to the number of tasks of the GP in question. The parameters $\sigma_f$, $l$, $\mathbf{B}$ and $\mathbf{v}$ are to be learned during the training process. For more examples of possible kernels see e.g. Rasmussen & Williams (2006); MacKay (1998). The Gram matrix is then constructed via the Kronecker product

$$\mathbf{K}_{\mathbf{f},\mathbf{f}'}(\mathbf{X}, \mathbf{X}') = k_{\text{RBF}}(\mathbf{X}, \mathbf{X}') \otimes \mathbf{k_t}. \tag{B.13}$$

We chose constant mean functions for all outputs of the multitask GP. All the models have been implemented in python with the library gpytorch (Gardner et al., 2018).

## B.6   Special Case of Constant Constraints

In Section 3.2 in the main paper, we detailed the method for incorporating the sum constraint into the GP in the general, non-constant case. Subsequently, in Section 3.2.1, we pointed out the possibility of implementing the sum constraint in a more efficient way for the case, where all of the constraints are constant and where the kernel of the GP factorizes into data and task kernel, as in (4) and (B.13). The main ideas are discussed in the main paper, here we summarize the modified procedure in Algorithm 3. A proof that the factorization holds also for the constrained GP is given in the next section.

### B.6.1   Proof of factorization

In this section we provide formal proof that the claims made in Section 3.2.1 hold, i.e. that directly constraining the mean and task covariance matrix and subsequently performing the Kronecker product with the data mean and covariance matrix does indeed lead to the constrained GP from Section 3.2.

To start, let us summarize the objects involved:

$$\boldsymbol{\Sigma}_{\text{tot}} = \mathbf{K} \otimes \boldsymbol{\Sigma_t} \tag{C.1}$$
$$\boldsymbol{\mu}_{\text{tot}} = \mathbf{m} \otimes \boldsymbol{\mu_t}$$
$$\mathbf{F}_{\text{tot}} = \mathbf{I}_{N_{\text{tot}}} \otimes \mathbf{F}$$
$$\mathbf{S}_{\text{tot}} = \mathbf{1}_{N_{\text{tot}}} \otimes \mathbf{S}$$

Here, $\mathbf{K}$ and $\boldsymbol{\Sigma_t}$ denote data and task covariance matrix, whereas $\mathbf{m}$ and $\boldsymbol{\mu_t}$ denote data and task mean, respectively. $\mathbf{F}$ and $\mathbf{S}$ define the constraint at a single point. $\mathbf{I}_{N_{\text{tot}}}$ denotes identity matrix and $\mathbf{1}_{N_{\text{tot}}}$ a vector of only ones of dimension $N_{\text{tot}}$. The quantities with the $_{\text{tot}}$ subscript give the quantities that correspond to the general approach from Section 3.2.

Due to the requirement of constant constraint and inter-task dependencies, we also need to pick a constant data mean $\mathbf{m}$, with entries $a = \text{const}$. We introduce the new quantity $\mathbf{S}' = \frac{\mathbf{S}}{a}$, which is used when constraining the task mean and covariance matrix. With equation (8), we find the following:

$$\mathbf{D}_{\text{tot}} = (\mathbf{F}_{\text{tot}} \boldsymbol{\Sigma}_{\text{tot}} \mathbf{F}_{\text{tot}}^{\mathsf{T}})^{-1} \mathbf{F}_{\text{tot}} \boldsymbol{\Sigma}_{\text{tot}}^{\mathsf{T}} \tag{C.2}$$
$$= \left( (\mathbf{I}_{N_{\text{tot}}} \otimes \mathbf{F})(\mathbf{K} \otimes \boldsymbol{\Sigma_t})(\mathbf{I}_{N_{\text{tot}}} \otimes \mathbf{F})^{\mathsf{T}} \right)^{-1} (\mathbf{I_N} \otimes \mathbf{F})(\mathbf{K} \otimes \boldsymbol{\Sigma_t})^{\mathsf{T}}$$
$$= \left( \mathbf{K} \otimes (\mathbf{F}\boldsymbol{\Sigma_t}\mathbf{F}^{\mathsf{T}}) \right)^{-1} (\mathbf{K}^{\mathsf{T}} \otimes \mathbf{F}\boldsymbol{\Sigma_t}^{\mathsf{T}})$$
$$= \left( \mathbf{K}^{-1} \otimes (\mathbf{F}\boldsymbol{\Sigma_t}\mathbf{F}^{\mathsf{T}})^{-1} \right) (\mathbf{K}^{\mathsf{T}} \otimes \mathbf{F}\boldsymbol{\Sigma_t}^{\mathsf{T}})$$
$$= \mathbf{K}^{-1}\mathbf{K}^{\mathsf{T}} \otimes (\mathbf{F}\boldsymbol{\Sigma_t}\mathbf{F}^{\mathsf{T}})^{-1}\mathbf{F}\boldsymbol{\Sigma_t}^{\mathsf{T}}$$
$$= \mathbf{I}_{N_{\text{tot}}} \otimes \mathbf{D}$$

$$\mathbf{A}_{\text{tot}} = \mathbf{I}_{N_{\text{tot}}} \otimes \mathbf{I_{N_f}} - \mathbf{D}_{\text{tot}}^{\mathsf{T}} \mathbf{F}_{\text{tot}} \tag{C.3}$$
$$= \mathbf{I}_{N_{\text{tot}}} \otimes \mathbf{I_{N_f}} - (\mathbf{I}_{N_{\text{tot}}} \otimes \mathbf{D})^{\mathsf{T}}(\mathbf{I}_{N_{\text{tot}}} \otimes \mathbf{F})$$
$$= \mathbf{I}_{N_{\text{tot}}} \otimes \mathbf{I_{N_f}} - \mathbf{I}_{N_{\text{tot}}} \otimes \mathbf{D}^{\mathsf{T}}\mathbf{F}$$
$$= \mathbf{I}_{N_{\text{tot}}} \otimes (\mathbf{I_{N_f}} - \mathbf{D}^{\mathsf{T}}\mathbf{F})$$
$$= \mathbf{I}_{N_{\text{tot}}} \otimes \mathbf{A}$$

$$\boldsymbol{\mu}'_{\text{tot}} = \mathbf{A}_{\text{tot}}\boldsymbol{\mu}_{\text{tot}} + \mathbf{D}_{\text{tot}}^{\mathsf{T}}\mathbf{S}_{\text{tot}} \tag{C.4}$$
$$= (\mathbf{I}_{N_{\text{tot}}} \otimes \mathbf{A})(\mathbf{m} \otimes \boldsymbol{\mu_t}) + (\mathbf{I}_{N_{\text{tot}}} \otimes \mathbf{D})^{\mathsf{T}}(\mathbf{1}_{N_{\text{tot}}} \otimes \mathbf{S})$$
$$= \mathbf{m} \otimes \mathbf{A}\boldsymbol{\mu_t} + \mathbf{1}_{N_{\text{tot}}} \otimes \mathbf{D}^{\mathsf{T}}\mathbf{S}$$
$$= \mathbf{m} \otimes (\mathbf{A}\boldsymbol{\mu_t} + \mathbf{D}^{\mathsf{T}}\mathbf{S}')$$
$$= \mathbf{m} \otimes \boldsymbol{\mu}'_t$$

$$\boldsymbol{\Sigma}'_{\text{tot}} = \mathbf{A}_{\text{tot}}\boldsymbol{\Sigma}_{\text{tot}}\mathbf{A}_{\text{tot}}^{\mathsf{T}} \tag{C.5}$$
$$= (\mathbf{I}_{N_{\text{tot}}} \otimes \mathbf{A})(\mathbf{K} \otimes \boldsymbol{\Sigma_t})(\mathbf{I}_{N_{\text{tot}}} \otimes \mathbf{A})^{\mathsf{T}}$$
$$= \mathbf{K} \otimes (\mathbf{A}\boldsymbol{\Sigma_t}\mathbf{A}^{\mathsf{T}})$$
$$= \mathbf{K} \otimes \boldsymbol{\Sigma}'_t$$

Hence we have shown that $\boldsymbol{\mu}'_{\text{tot}}$ and $\boldsymbol{\Sigma}'_{\text{tot}}$ of the constrained GP factorize into Kronecker products between the data mean and covariance matrix and the constrained task mean and covariance matrix, respectively.

### B.7 Credible Intervals

While standard deviation and variance for the backtransformed outputs $f$ cannot be recovered via a simple backtransformation of the corresponding quantities of $f'$, due to the potentially nonlinear and piecewise backtransformation, it is possible to recover credible intervals in this way: for the transformed outputs $f'$, we generate the upper and lower bounds of the $2\sigma$ credible interval; subsequently those bounds can be backtransformed in the same way as we do for the mean of the GP. That means that the posterior of the constrained $f$ can be a bit skewed, i.e. the mean may not lie exactly in the middle between upper and lower credible interval. When auxiliary outputs are involved in the backtransformation, their respective means should be used also when recovering the credible intervals, for the results to be consistent with the posterior means.

### B.8 Training Procedure

The models have been trained using the Adam optimizer provided by gpytorch. For each experiment, the corresponding learning rate (lr), number of iterations (iter) and (if applicable) scheduler settings are given in Table 7. The scheduler multiplies the learning rate with s-factor after s-steps iterations. The two different scheduler parameters given for the double pendulum correspond to the constrained and the unconstrained GP, respectively.

During the training of all datasets, we checked for errors in the Cholesky decomposition, which can happen when a matrix becomes singular due to numerical errors; when that happened, hyperparameter optimization was restarted with a new random initialization. For the non-square nonlinearity (logsin) experiment, training of the constrained GP proved to be less stable than for the other datasets. To counteract the issue, we tested for two further failure modes of the GP. First, we checked the learned lengthscale of the GP; if it was unreasonably small (smaller than 0.1), the training was repeated. Very small lengthscales typically correspond to the case where the GP learns an almost constant function with spikes towards all of the training points. Secondly, we confirmed that gradient descent had actually converged during training: to this end, we took the loss values over the last 40 iterations and checked, whether the standard deviation was smaller than 0.1. If either of the two checks failed, the training was repeated with newly initialized hyperparameters.

|                    | lr  | iter | s-steps | s-factor |
|--------------------|-----|------|---------|----------|
| HO (Sec. 4.1)      | 0.1 | 200  | 100     | 0.5      |
| Triangle (Sec. 4.2)| 0.1 | 2000 | 800     | 0.2      |
| DP (Sec. 4.3)      | 0.1 | 2000 | 800/500 | 0.2/0.5  |
| Free fall (A.1)    | 0.1 | 200  | 100     | 0.5      |
| Damped HO (A.2)    | 0.1 | 200  | 100     | 0.5      |
| Non-square (A.3)   | 0.1 | 200  | 100     | 0.5      |

Table 7: Training parameters for the experiments.

### B.9 Computing Power Available for the Experiments

All experiments have been conducted on a system with NVIDIA GTX 1060, 6GB GPU, Intel Core i7 7700-K @ 4.2GHz CPU and 16GB RAM. The creation of average values for one set of parameters as displayed in Tables 1-4 typically took between 15 minutes and three hours.

## C  Details on Simulated Datasets

In this section we provide information on how the data used in the different simulation experiments was generated.

### C.1 Harmonic Oscillator

The data for the harmonic oscillator toy problem was generated from

$$z(t) = z_0 \sin(\omega_0 t), \tag{C.1a}$$

$$v(t) = z_0 \omega_0 \cos(\omega_0 t). \tag{C.1b}$$

The energy is given by

$$E = \frac{k}{2} z(t)^2 + \frac{m}{2} v(t)^2 = \tag{C.2}$$

$$= \frac{k}{2} z_0^2 \sin^2(\omega_0 t) + \frac{m}{2} z_0^2 \omega_0^2 \cos^2(\omega_0 t) = \frac{k}{2} z_0^2. \tag{C.3}$$

We have chosen $E = 0.8\,\mathrm{J}$, $m = 1\,\mathrm{kg}$, $\omega_0 = 1\,\mathrm{s}^{-1}$ and it holds that $k = m\omega_0^2$ and $z_0 = \sqrt{2E/k}$.

Training data has been generated by evaluating the function on the equally spaced grid $t \in \mathrm{linspace}(0,10,20)$ [s]. Subsequently, random noise $\epsilon \sim \mathcal{N}(0, \sigma_n^2)$ was added to the data and output components were omitted at random with probability $f_d$; the values for $\sigma_n$ and $f_d$ are given in Table 1 in the main text. Test data has been generated on the grid $t \in \mathrm{linspace}(-0.1,10,100)$ [s].

### C.2 Damped Harmonic Oscillator

The data for the damped harmonic oscillator was generated from

$$z(t) = z_0'(t) \sin(\omega t), \tag{C.4a}$$

$$v(t) = z_0'(t)\omega \cos(\omega t) - z_0'(t)\frac{b}{2m} \sin(\omega t), \tag{C.4b}$$

where $z_0'(t) = z_0 \exp\left(\frac{-bt}{2m}\right)$ and $\omega = \sqrt{\omega_0^2 - \left(\frac{b}{2m}\right)^2}$. The energy is given by

$$E(t) = \frac{k}{2} z(t)^2 + \frac{m}{2} v(t)^2, \tag{C.5}$$

which is now time dependent and no longer yields a constant expression.

We have chosen $E = 0.8\,\mathrm{J}$, $m = 1\,\mathrm{kg}$, $\omega_0 = 1\,\mathrm{s}^{-1}$, $b = 0.1\,\mathrm{kg\,s}^{-1}$ and it holds that $k = m\omega_0^2$ and $z_0 = \sqrt{2E/k}$.

Training data has been generated by evaluating the function on the equally spaced grid $t \in \mathrm{linspace}(0,10,20)$ [s]. Subsequently, random noise $\epsilon \sim \mathcal{N}(0, \sigma_n^2)$ was added to the data and output components were omitted at random with probability $f_d$; the values for $\sigma_n$ and $f_d$ are given in Table 4. Test data has been generated on the grid $t \in \mathrm{linspace}(-0.1,10,100)$ [s].

### C.3 Free Fall

The data for the free fall was generated from

$$z(t) = v_0 t - \frac{g}{2} t^2, \tag{C.6a}$$

$$v(t) = v_0 - gt. \tag{C.6b}$$

The energy is given by

$$E = mgz(t) + \frac{m}{2} v(t)^2 = \frac{m}{2} v_0^2. \tag{C.7}$$

We have chosen $E = 200\,\mathrm{J}$, $m = 1\,\mathrm{kg}$ and it holds that $v_0 = \sqrt{2E/m}$, and the gravitational acceleration on earth is $g = 9.81\,\mathrm{m\,s}^{-2}$. Training data has been generated by evaluating the function on the equally spaced grid $t \in \mathrm{linspace}(0,6,20)$ [s]. Subsequently, random noise $\epsilon \sim \mathcal{N}(0, \sigma_n^2)$ was added to the data and output components were omitted at random with probability $f_d$; the values for $\sigma_n$ and $f_d$ are given in Table 3. To ensure good visibility and learnability, we scaled the data $\mathbf{y}$ with a factor $a = 20$: $\mathbf{y} \to \mathbf{y}/a$. Both in Figure 4 and in Table 3, the results are given in terms of the rescaled data (and noise values); the results in terms of the original scale can be obtained by multiplying with $a$. Test data has been generated on the grid $t \in \mathrm{linspace}(-0.1,6,100)$ [s].

## C.4 Non-square Nonlinearity

The data for the experiment with non-square nonlinearities was generated from

$$f_1(x) = 2e^{-5(x-1)^2} + e^{-5(x+1)^2} + 0.2, \tag{C.8}$$

$$f_2(x) = -\frac{x^3}{2}. \tag{C.9}$$

Training data has been generated on the equally spaced grid $x \in \text{linspace(-1.2,2,20)}$. Subsequently, random noise $\epsilon \sim \mathcal{N}(0, \sigma_n^2)$ was added to the data and output components were omitted at random with probability $f_d$; the values for $\sigma_n$ and $f_d$ are given in Table 5. Test data has been generated on the grid $t \in \text{linspace(-1.2,2,100)}$.

## C.5 Triangle in the Plane

In terms of the parameter $\alpha$, the trajectory that we used for the triangle in the plane in Section 4.2 is given by

$$\mathbf{Z}_0 = \begin{bmatrix} 4 & 8 & 8.4 \\ 4 & 4 & 6 \end{bmatrix}, \tag{C.10a}$$

$$\mathbf{Z}_1 = \mathbf{Z}_0 + d(\alpha), \tag{C.10b}$$

$$\mathbf{Z} = \mathbf{R}(\alpha)\mathbf{Z}_1 + d(\alpha), \tag{C.10c}$$

where each column of the matrix $\mathbf{Z}$ contains the coordinates of one corner point of the triangle, and where $d(\alpha) = \frac{1}{2}\cos(2\alpha)$ and $\mathbf{R}(\alpha)$ is a rotation matrix. Subsequently, random noise $\epsilon \sim \mathcal{N}(0, \sigma_n^2)$ was added to $\mathbf{Z}$; the values for $\sigma_n$ are given in Table 2 in the main text. We then added an auxiliary point of known position $(4, 4)$ to each datapoint, which will be important for the backtransformation:

$$\mathbf{Z} = \begin{bmatrix} z_{1x} & z_{2x} & z_{3x} & 4 \\ z_{1y} & z_{2y} & z_{3y} & 4 \end{bmatrix}. \tag{C.11}$$

Following the approach from Salzmann & Urtasun (2010a), we constructed the matrix $\mathbf{Q} = \mathbf{Z}^\mathsf{T}\mathbf{Z}$ and used the upper triangular elements of $\mathbf{Q}$ as transformed outputs for the constrained GP:

$$\mathbf{y}' = [Q_{11}, Q_{12}, Q_{13}, Q_{14}, Q_{22}, Q_{23}, Q_{24}, Q_{33}, Q_{34}, Q_{44}]. \tag{C.12}$$

Then the matrix $\mathbf{F}$ and the corresponding vector $\mathbf{S}$ encoding the length constraints for all the edges of the triangle become

$$\mathbf{F} = \begin{bmatrix} 1 & -2 & 0 & 0 & 1 & 0 & 0 & 0 & 0 & 0 \\ 1 & 0 & -2 & 0 & 0 & 0 & 0 & 1 & 0 & 0 \\ 0 & 0 & 0 & 0 & 1 & -2 & 0 & 1 & 0 & 0 \\ 0 & 0 & 0 & 0 & 0 & 0 & 0 & 0 & 0 & 1 \end{bmatrix}, \tag{C.13}$$

$$\mathbf{S} = \begin{bmatrix} L_{12}^2 & L_{13}^2 & L_{23}^2 & L_{04}^2 \end{bmatrix}^\mathsf{T}, \tag{C.14}$$

where $L_{ij}$ denote the distances between the points $i$ and $j$. The the last row of $\mathbf{F}$ corresponds to the constraint on the distance between the auxiliary point and the origin of the coordinate system. Note, that $Q_{14}$, $Q_{24}$ and $Q_{34}$ could in principle be learned separately from the remaining transformed outputs, as they do not enter into any of the constraints and the corresponding columns in (C.13) are zero. Furthermore, $Q_{44}$ could be omitted from the learning process entirely, as the value is known.

After training the constrained GP, the predicted values $\mathbf{f}'$ are rearranged into the (symmetric) matrix $\widetilde{\mathbf{Q}}$, analogously to (C.12). Then the matrix $\widetilde{\mathbf{Z}}$ is recovered via a singular value decomposition (SVD) of $\widetilde{\mathbf{Q}}$. This decomposition is not unique and the auxiliary point comes into play: we compare the learned with the known position and determine the angle between them, which enables us to rotate the learned coordinates to their true positions.

Training data was generated on the grid $\alpha \in [0, 5]$, consisting of 20 uniformly spaced points. Subsequently, random noise $\epsilon \sim \mathcal{N}(0, \sigma_n^2)$ was added to the data; the values for $\sigma_n$ are given in the main text. Test data was generated over the same range $[0, 5]$, although this time with the grid divided into 100 points.

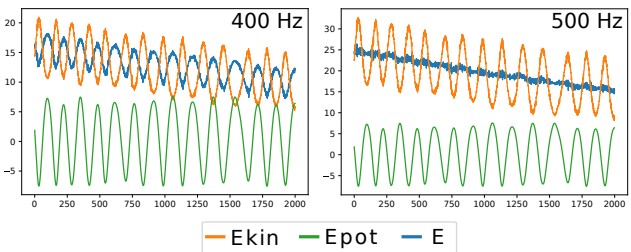

Figure 8: Comparison of the energy of the double pendulum for different frame rates of the camera. It is apparent that a frame rate of 400 Hz is incompatible with the principle of energy conservation; while the energy is decreasing in the long term due to friction, it should never increase. No choice of mass ratio $m_b/m_g$ was able to resolve this issue. On the other hand, a frame rate of 500 Hz together with the mass ratio $m_b/m_g = 6.5$ is compatible with energy conservation, within the bounds of error.

## D  Details on the Double Pendulum Dataset

### D.1  Parameters

In Section 4.3 we demonstrated the applicability of our approach to the 'Double Pendulum Chaotic' dataset. A description of the dataset can be found in Asseman et al. (2018); to prevent confusion, we should mention that the blue and the green marker in our paper correspond to the green and the blue marker in Asseman et al. (2018), respectively (i.e. the colors have been exchanged). The lengths of the two pendula are given as $l_b = 91$ mm and $l_g = 70$ mm, where the subfix $b$ refers to the pendulum with blue marker and $g$ to the one with green marker. However, in order to calculate the energy (up to a constant factor), knowledge of the masses, or at least of the ratio $m_b/m_g$ is required. From information given by the authors of the paper and the manufacturer of the double pendulum, together with some experimentation of our own we estimated this ratio as $m_b/m_g \approx 6.5$. Note that in our description of the double pendulum (11), we made the assumption that it consists of two point masses, which is only approximately true.

Another quantity of interest is the frame rate of the camera that was used to create the dataset; it enters into the model when calculating the velocities of the masses. In their paper (Asseman et al., 2018), the authors state a frame rate of 400 Hz. However, our experiments with the dataset and keeping the energy constraint in mind strongly indicate a frame rate of 500 Hz; for 400 Hz there are segments of the motion where the total energy $E$ clearly increases which violates the principle of energy conservation (see Figure 8).

The 'Double Pendulum Chaotic' dataset was published under the "Community Data License Agreement - Sharing - Version 1.0".

### D.2  Implementation Details

The 'Double Pendulum Chaotic' dataset provides data in the form of annotated positions of the masses attached to the ends of the two pendula (together with the position of the top of the apparature holding the pendulum which does not change and which we therefore omitted). We now have the positions as points on an equally spaced grid; in terms of the camera frame rate $r$ the spacing between two adjacent points is given by $1/r$. To obtain the velocities we numerically take the gradient of the positions on the grid and we receive the data which we use for our GP, with outputs

$$\mathbf{f} = [z_{bx}, z_{by}, z_{gx}, z_{gy}, v_{bx}, v_{by}, v_{gx}, v_{gy}]^\mathsf{T}. \tag{D.1}$$

To obtain positions and velocities with comparable absolute values, which enhances the performance of the GP and which makes the quantities easier to compare in plots, we scaled positions by a factor of 20 and velocities by a factor of $\sqrt{10}$; the time $t$ was scaled by a factor of 5.

As outlined in Section 4.3, we obtain training data, to be used during hyperparameter optimization, and test data, to evaluate the quality of predictions, by picking a random interval of 200 datapoints from the

second half of the trajectories provided by the dataset; out of those we use 15 points as training data and the rest as test data. Note that the value $\hat{E}$ received by evaluating (11) and averaging over the training data will in general be a less accurate estimate than the value of the energy $\hat{E}_0$ received when averaging over all datapoints in the interval, since the average is taken over fewer points in the former case. Hence, when determining the accuracy of the constraint fulfillment, the results in Section 4.3 have been compared to $\hat{E}_0$.

For the double pendulum, we receive the transformed outputs

$$\mathbf{f}' = [z_{by}, z_{gy}, v_{bx}^2, v_{by}^2, v_{gx}^2, v_{gy}^2]^\mathsf{T}, \tag{D.2}$$

with corresponding

$$\mathbf{F} = [m_b g, m_g g, \frac{m_b}{2}, \frac{m_b}{2}, \frac{m_g}{2}, \frac{m_g}{2}]. \tag{D.3}$$

The auxiliary outputs are

$$\mathbf{f}_{\text{aux}} = [z_{bx}, z_{gx}, v_{bx}, v_{by}, v_{gx}, v_{gy}]^\mathsf{T}; \tag{D.4}$$

note, that the outputs $z_{bx}$ and $z_{gx}$ are not actually auxiliary outputs, but since they are not involved in the constraint (11) (i.e. the corresponding entry in $\mathbf{F}$ would be zero), they can be learned separately from the constrained outputs, together with the auxiliary outputs. Same as for the harmonic oscillator, we created virtual measurements for $v_{bx}^2, v_{by}^2, v_{gx}^2, v_{gy}^2$ at zero crossings of the auxiliary outputs $v_{bx}, v_{by}, v_{gx}, v_{gy}$.

# E  Comparison to Jidling et al. (2017)

In this section we will investigate the parallels between the method of Jidling et al. (2017) and our own. While they only consider homogeneous constraints in the paper, in the context of constant linear sum constraints it is simple to extend the method to affine constraints, as we will see below.

In the approach of Jidling et al. (2017), vectors spanning the nullspace of the constraint $\mathcal{F}$ are used to construct the task covariance matrix. Given a sum constraint $\mathcal{F}(\mathbf{f}) = \mathbf{F}\mathbf{f} = \mathbf{S}$, and vectors $\mathbf{h}_i$ spanning the nullspace (i.e. $\mathbf{F}\mathbf{h}_i = 0$), we can define the matrix

$$\mathbf{G} = [\mathbf{h}_1 \mathbf{h}_2 \dots \mathbf{h}_n], \tag{E.1}$$

where $n$ denotes the dimension of the nullspace. A suitable task covariance matrix can then be constructed via $\mathbf{k_t} = \mathbf{G}\mathbf{G}^\mathsf{T}$, and in order to accommodate the non-zero right hand side of the sum constraint, the task mean $\mathbf{m_t}$ is chosen such that $\mathbf{F}\mathbf{m_t} = \mathbf{S}$. Then we obtain the multivariate Gaussian $\mathcal{N}(\mathbf{m_t}, \mathbf{k_t})$, samples of which obey the constraint $\mathcal{F}$. Note that $\mathbf{k_t}$ is the projector on the nullspace of the constraint; in Matthews et al. (2017), the relationship between constrained multivariate Gaussian distributions and the nullspace of the corresponding linear operator is discussed. Subsequently, the full mean and covariance matrix of the GP can be constructed according to (4).

To make this more concrete, we consider again the example of the harmonic oscillator from Section 3.1.2. Here, the constraint is given by $\mathbf{F} = [k/2, m/2, 0, 0]$ and $\mathbf{S} = E$. Then the corresponding matrix $\mathbf{G}$ can be constructed as (the choice of null vectors is not unique)

$$\mathbf{G} = \begin{bmatrix} \frac{m}{\sqrt{m^2+k^2}} & 0 & 0 \\ \frac{-k}{\sqrt{m^2+k^2}} & 0 & 0 \\ 0 & 1 & 0 \\ 0 & 0 & 1 \end{bmatrix}. \tag{E.2}$$

The task mean and task covariance matrix become

$$\mathbf{k_t} = \mathbf{G}\mathbf{G}^\mathsf{T} = \begin{bmatrix} \frac{m^2}{m^2+k^2} & \frac{-mk}{m^2+k^2} & 0 & 0 \\ \frac{-mk}{m^2+k^2} & \frac{k^2}{m^2+k^2} & 0 & 0 \\ 0 & 0 & 1 & 0 \\ 0 & 0 & 0 & 1 \end{bmatrix}, \qquad \mathbf{m_t} = \begin{bmatrix} \frac{E}{k} \\ \frac{E}{m} \\ 0 \\ 0 \end{bmatrix}. \tag{E.3}$$

Now if we approach the problem from the other side, as it turns out, starting with the identity matrix as task covariance matrix and then conditioning it according to (8) leads to the same $\mathbf{k_t}$ obtained in (E.3). Here we note one advantage of our approach: it is straightforward to include additional correlations into the task covariance matrix, e.g. correlations between constrained outputs and those not involved in the constraint.

For example, when introducing an additional correlation between tasks one and three we obtain (after setting $m = k = 1$)

$$\mathbf{k_t^0} = \begin{bmatrix} 1 & 0 & 0.5 & 0 \\ 0 & 1 & 0 & 0 \\ 0.5 & 0 & 1 & 0 \\ 0 & 0 & 0 & 1 \end{bmatrix} \longrightarrow \mathbf{k_t} = \begin{bmatrix} 0.5 & -0.5 & 0.25 & 0 \\ -0.5 & 0.5 & -0.25 & 0 \\ 0.25 & -0.25 & 0.875 & 0 \\ 0 & 0 & 0 & 1 \end{bmatrix}, \tag{E.4}$$

where $\mathbf{k_t^0}$ and $\mathbf{k_t}$ are the unconstrained and the constrained task covariance matrix, respectively. So the constraint alters correlations between outputs involved in the constraint and other outputs. In the approach of Jidling et al. (2017), correlations between the nullspace dimensions could be added by introducing a non-diagonal matrix between $\mathbf{G^T}$ and $\mathbf{G}$ in (E.3). However, that method would not allow us to introduce arbitrary correlations between the tasks as demonstrated in (E.4). Throughout the work, we have used the structure given in (B.12) as the starting point for our task covariance matrices.

