# OpenReview forum: "Incorporating Sum Constraints into Multitask Gaussian Processes"
_TMLR — Accepted by TMLR_

### Review · Reviewer_aAJV · 2022-09-08

**Summary Of Contributions:**

This work tackles the problem of leveraging prior knowledge regarding sum constraints in multi-task gaussian process (GP) regression. These problems arise in physics domains, where often multiple possibly correlated outputs are known to obey sum constraints. This paper demonstrates how sum constraints that are non-linear functions of the outputs can be transformed into linear sum constraints by transforming the outputs. The paper then provides a procedure for fitting multi-task GPs under sum constraints and using them for prediction. The method is empirically evaluated on two synthetic and one real data example, where the proposed approach achieves lower constraint violation on out of sample prediction tasks than multi-task GPs that do not account for constraints and comparable or better RMSE.

**Requested Changes:**

Important
- In Sec 3.2.1, how are the constraints $F$ and $S$ changed when they are only applied to the inter-task covariance matrix directly?
- It is unclear at several points in the paper if the setting considered is where all outputs are observed for each input. In section 2.1, a Kronecker product $\otimes$ is used (as in 3.2.1), but it is only a Kronecker product if all outputs are observed for each input. In the harmonic oscillator example in figure 1, not all outputs are observed for each input. Hence, the multi-task kernel should be a Hadamard product, not a Kronecker product. Please unify and clarify this in the paper.

Strengthen:

- Is the auxiliary variable always simply $f$? Are there examples where other auxiliary variables should be used?
- How are the inputs corresponding to virtual observations obtained? Are they found by finding where the posterior mean is zero?
- In figure 1, it would be nice to show the unconstrained GP after applying the transformation to its outputs (e.g. by Monte-Carlo sampling in cases where analytic representation is not tractable).
- What are the plus or minus terms in the tables? Are these numbers two standard errors of the mean? Please state this.
- Sec 4.3, what is the training data (how many points and from which trajectories) for the double pendulum problem?
- A metric that evaluates the quality of the uncertainty estimation such as log-likelihood would be nice to add.

Minor Grammar:
- There are a couple instances where hyphens are used instead of em dashes (e.g. Sec 2.1).
- Sec 2.1: “and observations” -> and observation vector
- Sec 4: missing space after period
- Sec B2: Shouldn’t $\boldsymbol f’$ be used instead of $\boldsymbol f$ throughout this section?
- Using $N_\text{tot}$ for the number of training points and the number of training + test points is confusing
- Figure 3: the right subplot is confusing. Is the left inset the unconstrained GP and the right inset the constrained GP? The caption is a little confusing, so a clearer description or legend would be helpful

**Strengths And Weaknesses:**

Strengths

- Incorporating prior knowledge in probabilistic modeling is relevant and of interest to the TMLR community.
- The paper is easy-to-follow and well-written.  The paper uses a harmonic oscillator example throughout the text, which provides a nice demonstration of the ideas.
- The proposed constrained multi-task GP outperforms alternatives in terms of constraint violations, while achieving comparable or better RMSE
- Some limitations are discussed (discontinuities, potentially for overfitting due to virtual observations)

Weaknesses

- Depending on the transformation, the transformed outputs may not be well modeled with a standard stationary GP kernel. An RBF kernel is used in all experiments (for constrained and unconstrained GPs), but it seems unlikely that this is the ideal choice for both. To what extent is the choice of kernel impactful on this evaluation?
- The Laplace approximation requires that the likelihood of the transformed observations be known. How common is this expected to be? Arbitrary non-linear transformations of a Gaussian distribution would not follow well known probability distributions. Is the proposed method applicable in such cases?
 - The quality of the Laplace approximation (i.e. the ability to approximate the likelihood with a Gaussian) will depend on the transformation. The Laplace approximation is used in all examples in this paper. The authors note that the approximation should be evaluated on a case by case basis. Is the proposed technique amenable to alternative approximations if the Laplace approximation is poor?
- For arbitrary non-linear constraints, it seems that appropriate transformations may not be obvious. Are there cases where such transformation is not possible or consist of a very large pieces?

---

> ### Author Response · Authors · 2022-09-24
> **Reply to reviewer aAJV**
>
> We thank the reviewer for the careful evaluation of our paper. To address the comments listed as weaknesses:
>
> Ad 1: We would like to stress that both GPs are learned independently and are thus not restricted to use the same GP prior. While different kernels could be used, the RBF kernel performed well for the examples considered in the paper.
>
> Ad 2: The formula given in (B.8) gives a way of determining the likelihood, also in cases where it does not coincide with well known probability distributions.
>
> Ad 3: The proposed method is amenable to alternative approximative techniques, such as variational inference or expectation propagation. The equations in (B.7) will then need to be replaced by expressions corresponding to these techniques. We have added a brief statement on this matter in the last paragraph of Section 3.1.1. We would like to add that we did not use the Laplace approximation on all examples in the paper; we have now made this clearer at the end of Appendix B.3.
>
> Ad 4: In the paper we focus on constraints of the form (6), with known h_i and hence we do not claim to treat arbitrary non-linear constraints. Some non-linear constraints which are not of the form (6), such as the length-constraint (10), can still be enforced by the sum constraint methodology via a suitable transformation. In such cases, the transformations may not be obvious and individual solution strategies will need to be found. Even though we cannot treat arbitrary non-linear constraints, we consider the class of non-linear sum constraints which we esteem to be broad.
>
> To address the requested changes:
>
> Important:
>
> Ad 1: When the constraint is applied to the inter-task covariance matrix directly, F and S can be used directly in the conditioning. We have now clarified this in the text in Section 3.2.1 and in Algorithm 3.
>
> Ad 2: Thank you for pointing this out. To keep the method simple and to avoid notational clutter, we chose the following procedure: we treat incomplete measurements by first constructing the full matrices via the Kronecker product and subsequently removing the entries corresponding to missing measurements. In the paper, we now clarified this issue in the additional appendix B.2 and a clarifying statement at the end of Section 2.1. In addition, we added another Step to Algorithms 2 and 3 in order to state this explicitly.
>
>
> Strengthen:
>
> Ad 1: The auxiliary variable is not necessarily always f. Although for the examples considered it turned out to be a practical choice. We added a brief comment to the second paragraph of Section 3.1.2.
>
> Ad 2: As you say, they are obtained by determining the zero crossings of the corresponding posterior means. We have now made this clearer in the text in the last paragraph of Section 3.1.2 and the analogous passages of the other experiments.
>
> Ad 3: While it could easily be done, we feel that adding those lines to the middle plot would come at the cost of clarity. Since subplot 1 and 3 next to each other already allow for a direct comparison between the constrained and unconstrained GP, we chose to keep the plot as is.
>
> Ad 4: The plus-or-minus terms in the tables correspond to one standard deviation of the mean. We have now made this clear in the table captions.
>
> Ad 5: We believe that this information is already given in the first two paragraphs of Section 4.3.
>
> Ad 6: We decided against adding the the log-likelihood in the tables since the constrained and unconstrained GP are trained on different (transformed and non-transformed) data, making the values not directly comparable.
>
>
> Minor Grammar:
>
> We would like to thank the reviewer for pointing out these minor issues and we corrected them in the revised version of the paper. We chose to keep the the notion of N_tot as the alternatives would result in significantly more notational clutter.

---

> ### Author Response · Authors · 2022-10-05
> **Reply 2 to reviewer aAJV**
>
> To address your question on whether the method is amenable to alternative approximations other than the Laplace approximation, we also implemented variational inference and added Appendices A.4 and B.4 in the second revised version of the paper. While the Laplace approximation is superior in this case, it demonstrates that the method is in principle compatible with alternative approximations.

---

> > ### Comment · Reviewer_aAJV · 2022-10-19
> > **Response**
> >
> > Thanks for the clarifications and additions.

---

### Review · Reviewer_JvJ3 · 2022-09-13

**Summary Of Contributions:**

The authors consider the problem of inference using vector-valued Gaussian processes when certain equality constraints of the GP must be satisfied. These are, in order of difficulty, linear constraints (where the linear combination of a GP must be equal to some value), monotonic nonlinear constraints (where the linear combination of a monotonic nonlinear function of a GP is equal to some value), and arbitrary nonlinear constraints. Linear constraints are easily handled by the fact that evaluations of linear operators of Gaussian random variables are jointly Gaussian. Monotonic nonlinearities are handled via a Laplace approximation, and the underlying GP can be found by inverting the monotonic transformation. Arbitrary nonlinearities are handled in a similar way, but a pseudo-inverse is sought rather than an inverse. Three tasks are considered: a synthetic harmonic oscillator, a synthetic pose estimation problem, and a real double pendulum.

**Broader Impact Concerns:**

None.

**Requested Changes:**

As the authors mention in the Appendix,
``There is no guarantee that Newton’s method will determine the correct maximum ˆf in case of multimodal
distributions, or that the resulting Gaussian distribution will constitute a good approximation of the true
posterior. For these reasons, it has to be decided on a case by case basis whether the Laplace approximation
should be employed or not"

Perhaps I missed it, but I did not see anywhere in the examples a discussion on whether the Laplace approximation should be employed or not. Can you provide one? What are the advantages of Laplace approximation over other methods? Is it possible to try other methods for at least one of the problems you tried?



**Strengths And Weaknesses:**

Strengths:
- The method is principled and appears to have a sensible derivation. While perhaps obvious, it is nice to see it written down all in one place so that people can refer to the details.

Weaknesses
- The various scenarios under which GP posteriors can be inferred from observations of linear transformations of GPs is well-known, and follows directly from the basic properties of finite-dimensional Gaussian vectors. The ``multi-task" element does not really add much to the difficulty of the problem. The Laplace approximation is also well-known, as are its limitations. I don't think any reader will find the results here surprising or particularly enlightening, as the result could be conveyed with the summary I wrote above.

All-in-all, I am leaning very slightly towards acceptance, because while the results might not be surprising, it is nice to see all the details collected together in one place.

---

> ### Author Response · Authors · 2022-09-24
> **Reply to reviewer JvJ3**
>
> We thank the reviewer for the positive evaluation of our paper.
>
> To address the stated weaknesses:
>
> We would like to emphasize that the nonlinear sum constraint has not been discussed in the existing GP literature and that the proposed method may therefore provide new insights to many readers.
>
> To address the requested changes:
>
> The main advantage of using the Laplace approximation lies in its simplicity. In the last paragraph of Appendix B.3, we listed the examples for which the Laplace approximation was used. We ended up using standard GP regression over the Laplace approximation for the double pendulum and the triangle in the plane. Visual inspection of the GP outputs typically gives a good idea on whether the Laplace approximation performs well or breaks down. We added clarifying statements to the last paragraph of Appendix B.3.
>
> In principle, it would be possible to use e.g. variational inference or expectation propagation instead of the Laplace approximation. However, evaluating the performance of different approximation schemes is not the purpose of this paper. We have now added a comment to the last paragraph of Section 3.1.1 to clarify this.

---

> ### Author Response · Authors · 2022-10-05
> **Reply 2 to reviewer JvJ3**
>
> To address your request of trying out an alternative method of appxoimate inference, we also implemented variational inference and added Appendices A.4 and B.4 in the second revised version. From the example considered, it appears that the Laplace approximation is less prone to overfitting than the variational approach.

---

> > ### Comment · Reviewer_JvJ3 · 2022-10-06
> > **Thanks for your response**
> >
> > Thanks for your response and updates.

---

### Review · Reviewer_68RS · 2022-09-21

**Summary Of Contributions:**

This work propose a method to impose linear and nonlinear constraints on multiple outputs of a Gaussian process. It is based on the framework of multitask GPs, which decomposes the kernel into a kronecker product of task kernel and data kernel. Nonlinear constraints are transformed into linear ones through reparameterization. In order to address non-monotonic nonlinearities, an unconstrained GP is fit to the data to determine the region of the outputs. The paper also proposes to regularize the constrained GP with results from the unconstrained GP. Results on several toy and real-world datasets show that incorporating constraints often improves the prediction accuracy and reduce the credible intervals.

**Requested Changes:**

I would like to see my comment on the two claims addressed. And I also believe the artifact problem is a big concern and should either be solved or by providing evidence that it does not affect practical utility of the approach.

**Strengths And Weaknesses:**

## Strength

* This paper addresses an important problem of Gaussian process modelling, especially in the multi-output setting. As demonstrated by examples from physics, vision, and control domains, real-world modelling tasks often require imposing equality constraints over multiple outputs of a function. Efficient inference schemes must be developed in order to use GPs as priors for such functions. Therefore, work that addresses this problem will likely be of interest to a wide community.

* The idea is simple (reparameterizing nonlinear constraints as linear ones and fit gps to the transformed function values). It works well together with a clever trick that solves non-monotonic nonlinearity.

* The method was tested using a variety of experiments in real-world scenarios.

## Weaknesses

* I found several claims in the paper clueless:
   * Page 5, "In case of a position dependent constraint, it is important to note that the values of the functions c(x) and a(x) must be known at all N_{tot} points." I don't think this is necessary. Correct me if I am wrong, but I believe it is possible to only condition the GP on the constraints over some subset of points.
   * Section 3.2.1, "It now suffices to enforce the constraints F on the task mean and covariance matrix". I don't see how the constraints on the original GP translate to constraints on the task mean and covariance matrix. A proof should be given here if it exists. This is a major concern I have on the correctness of the proposed method.

* The complexity of the proposed method seems high (impractical). How does it apply to large datasets? Could it be combined with sparse variational inference approaches?

* There is artifacts in the uncertainty due to GP values might fall outside of the output domain of the variable (e.g., v^2 must be non-negative). This seems a major drawback of the proposed method.

---

> ### Author Response · Authors · 2022-09-24
> **Reply to reviewer 68RS**
>
> We thank the reviewer for the careful evaluation of our paper.
>
> To address the stated weaknesses:
>
> Ad 1 – Page 5: Indeed, it would also be possible to condition the GP over only a subset of points. As a consequence, the constraint would also be enforced at only this subset of points. We added a brief remark to the main text in Section 3.2 to clarify this.
>
> Ad 1 -  Section 3.2.1: We now provide a detailed proof in Appendix B.5.1.
>
> Ad 2: In principle, it should also be possible to combine the method with sparse variational inference methods. This constitutes an interesting avenue for future research. We have added a remark to the 'Conclusions and Future Work' section.
>
> Ad 3: Indeed, it can occasionally happen that the GP produces invalid values for the outputs. In Section 3.1.2, we explain how this issue can be dealt with heuristically. As the experimental results given in the paper demonstrate, the occurance of such outputs is typically limited to a small area, where the backtransformation switches from one local inverse to another. Their impact on the performance of our method is insignificant.

---

### Decision · Action_Editors · 2022-11-25

**Recommendation:** Accept as is

**Comment:**

Almost all questions/concerns raised during the discussion phase were addressed and clarified in the latest version. The reviewers noted that only a restricted class of non-linear constraints has been considered, the occasional invalid outputs produced by the approximation and the ideas seem to be straightforward. However, based on the paper's correctness and relevance to the TMLR community, an acceptance decision is recommended.

**Audience:**

This paper should be of interest to the probabilistic modelling / Gaussian process community,

**Claims And Evidence:**

The claims are supported by experiments and comparison to techniques that do not enforce constraints.